# RLVE: Scaling Up Reinforcement Learning for Language Models with Adaptive Verifiable Environments

**Zhiyuan Zeng** [* 1 2]  **Hamish Ivison** [* 1 2]  **Yiping Wang** [* 1]  **Lifan Yuan** [* 3]
**Shuyue Stella Li** [1]  **Zhuorui Ye** [4]  **Siting Li** [1]  **Jacqueline He** [1]  **Runlong Zhou** [1]  **Tong Chen** [1]  **Chenyang Zhao** [5]
**Yulia Tsvetkov** [1]  **Simon Shaolei Du** [1]  **Natasha Jaques** [1]  **Hao Peng** [3]  **Pang Wei Koh** [1 2]  **Hannaneh Hajishirzi** [1 2]

## Abstract

We introduce **R**einforcement **L**earning (RL) with Adaptive **V**erifiable **E**nvironments (**RLVE**), an approach using verifiable environments that procedurally generate problems and provide algorithmically verifiable rewards, to scale up RL for language models (LMs). RLVE enables each verifiable environment to dynamically adapt its problem difficulty distribution to the policy model's capabilities as training progresses. In contrast, static data distributions often lead to vanishing learning signals when problems are either too easy or too hard for the policy. To implement RLVE, we create **RLVE-GYM**, a large-scale suite of 400 verifiable environments carefully developed through manual environment engineering. Using RLVE-GYM, we show that environment scaling, i.e., expanding the collection of training environments, consistently improves generalizable reasoning capabilities. RLVE with joint training across all 400 environments in RLVE-GYM yields a 3.37% absolute average improvement across six reasoning benchmarks, starting from one of the strongest 1.5B reasoning LMs. By comparison, continuing this LM's original RL training yields only a 0.49% average absolute gain despite using over $3\times$ more compute. We release our code publicly.[1]

## 1. Introduction

Scaling up reinforcement learning (RL) has shown strong potential to improve language models (LMs) (Ouyang et al., 2022; OpenAI, 2024; DeepSeek-AI, 2025; Google Deep-Mind, 2025), but models' improvement increasingly saturates on finite training data (Kumar et al., 2024; Hu et al., 2025b; Khatri et al., 2025). Scaling up RL data presents two challenges. First, collecting a large number of problems along with their ground-truth answers, which are commonly required for verifiable reward computations (Lambert et al., 2025; DeepSeek-AI, 2025), can be expensive. Second, RL training can completely stall when problems are either too easy or too difficult for the policy model (Razin et al., 2024; 2025), since too easy ones provide no meaningful learning signal, whereas overly difficult ones yield consistently poor rewards that impede gradient-based updates. In typical LM RL training, the problem distribution is predetermined by a specific dataset and remains static, preventing adaptation to the policy model's evolving capabilities (Figure 1(a)).

To address these challenges, we introduce **RLVE** (Reinforcement Learning with Adaptive Verifiable Environments), an approach that scales up LM RL training by using verifiable environments whose difficulty dynamically adapts to the policy model's capabilities (Figure 1(b)). A verifiable environment (1) procedurally generates an unbounded number of problems with configurable difficulty, and (2) provides algorithmically verifiable rewards to model outputs. For example, an environment for an array-sorting task can sample an array and verify an output by comparing it with the result produced by a sorting program, overcoming the non-scalability of collecting problems individually; we can increase the difficulty by increasing the array length.

RLVE enables **adaptive** verifiable environments, in which the difficulty distribution shifts toward harder problems once the policy model performs well on the current distribution. RLVE thus addresses the limitations of static environments, whose distribution of generated problems remains constant throughout training. Empirically, after the model masters the hardest problems, the static environment becomes uninformative for further improvement; even if the model does not ultimately achieve this within a limited compute budget, learning efficiency remains suboptimal when the difficulty of most problems is inappropriate for the policy model. By

---

[*]Equal contribution  [1]University of Washington  [2]Allen Institute for Artificial Intelligence  [3]University of Illinois Urbana-Champaign  [4]Princeton University  [5]LMSYS Org. Correspondence to: Zhiyuan Zeng <zyzeng@cs.washington.edu>.

*Proceedings of the 43rd International Conference on Machine Learning*, Seoul, South Korea. PMLR 306, 2026. Copyright 2026 by the author(s).

[1]https://github.com/Zhiyuan-Zeng/RLVE

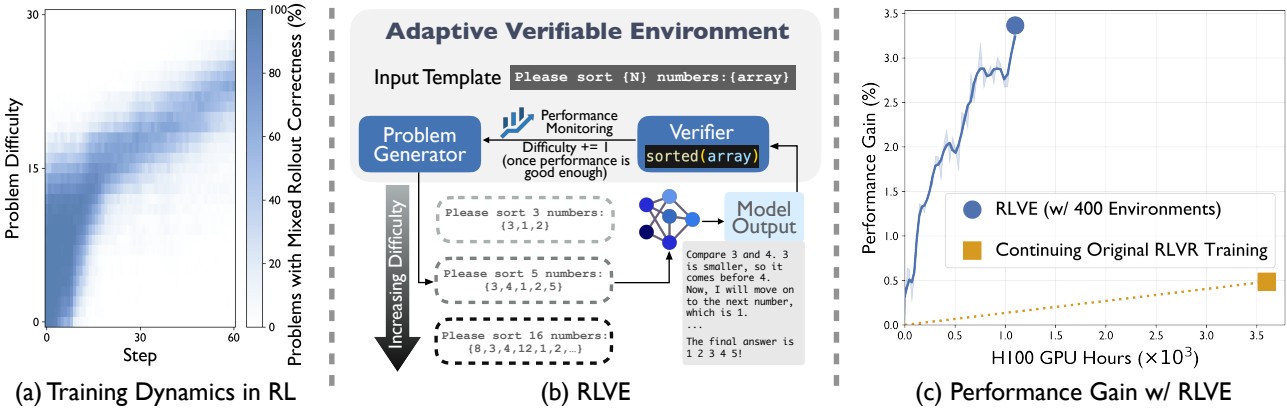

*Figure 1.* (a) During RL training, some array-sorting problems that were appropriately challenging become too easy, while others that were too hard become learnable as the policy improves (given the upward movement of the dark region containing many problems for which some rollouts are correct, and others are not). (b) RLVE trains an LM on verifiable environments that dynamically adjust problem difficulty. (c) Starting from `ProRL-1.5B-v2`, continuing training with RLVE yields a 3.37% absolute average improvement across six reasoning benchmarks, whereas continuing the original RLVR training achieves a 0.49% gain using more than 3× the compute.

contrast, our experiments in Section 4.1 show that adaptive difficulty continuously maintains a high proportion of appropriately challenging problems, leading to superior performance and higher learning efficiency.

To implement RLVE, we construct **RLVE-GYM**, a large-scale suite of 400 verifiable environments developed through our expert **environment engineering** efforts, following two principles. First, the environments are designed as pedagogical tools for developing reasoning capabilities. This procedure is analogous to teaching a pupil to perform integer multiplication by hand, although using a calculator is more efficient and reliable. Second, the environments enable output verification via two advantages: (1) the environment can execute programs while the LM is not allowed to do so, and (2) the environment is responsible only for verifying outputs rather than solving the problems, and verification is sometimes much easier than solving in terms of computational complexity, e.g., in NP-complete problems. Using RLVE-GYM, Section 4.2 shows that **environment scaling**, i.e., expanding the training collection of verifiable environments, consistently improves the model performance on environments unseen during training, underscoring its importance for developing generalizable reasoning capabilities.

We show that RLVE with joint training across all 400 environments in RLVE-GYM effectively scales up LM RL training in two scenarios, as reflected by performance on six reasoning benchmarks covering mathematics, code generation, and logical reasoning. The first is a data-saturation scenario. Section 5.1 demonstrates that RLVE further scales up RL training for one of the current strongest 1.5B RL LMs, `ProRL-1.5B-v2` (Hu et al., 2025a) (Figure 1(c)). This LM was originally trained with RLVR (RL with verifiable rewards) for over 20,000 H100 GPU hours to saturation on ProRL, a large-scale, diverse dataset (Liu et al., 2025e).

The second is a compute-constrained scenario, where training starts from an LM that has not undergone reasoning RL. Section 5.2 shows that RLVE outperforms training on a strong RLVR dataset, DeepMath-103K (He et al., 2025), by about 2% in absolute improvement, when both initialized from `OpenThinker3-1.5B` (Guha et al., 2025), one of the current strongest 1.5B SFT LMs, and following an identical training setup. Notably, RLVE requires no benchmark-specific data, whereas DeepMath-103K was explicitly designed for mathematical reasoning. In addition, constructing RLVE-GYM is substantially more cost-efficient, as DeepMath-103K required roughly $138,000 USD and 127,000 GPU hours to build (He et al., 2025).

We call on the community to advance research on adaptive environments, where data collection is inherently scalable and the difficulty is unbounded, with supervision signals continuously adapting right at the model's capability frontier. We believe that environment engineering will become as foundational to LM development as feature, data, and prompt engineering, and that this work is part of a broader effort to scale RL training through adaptive environments.

## 2. Methodology

We introduce **RLVE** (**R**einforcement **L**earning with Adaptive **V**erifiable **E**nvironments), an approach for scaling up LM RL training using procedurally generated data from verifiable environments, as defined in Section 2.1. RLVE dynamically generates problems from these environments during training and can be paired with any RL algorithm that uses environment-supplied rewards. Importantly, RLVE makes each environment adaptive, adjusting its problem difficulty distribution based on the evolving capabilities of the trained policy model, as detailed in Section 2.2.

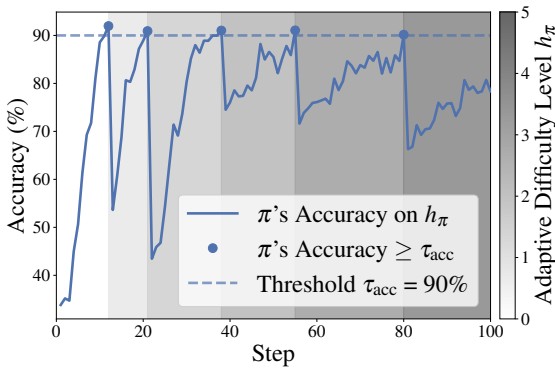

*Figure 2.* Illustration of adaptive difficulty enabled by RLVE when training a policy model $\pi$ on the `Sorting` environment. Shown are the adaptive difficulty level $h_\pi$ and the model $\pi$'s accuracy on problems generated from this level at each step. Whenever the accuracy exceeds the threshold $\tau_{\mathrm{acc}}$ (90%), RLVE increments $h_\pi$ by 1, shifting the difficulty distribution to harder problems.

## 2.1. Verifiable Environment

We define a *verifiable environment* as a tuple $E = (I, \mathcal{P}, R)$, where $I$ is an *input template*, $\mathcal{P}$ is a *problem generator*, and $R$ is a *verifier (reward function)*. The problem generator $\mathcal{P}$ procedurally samples problems that instantiate the input template $I$ to produce inputs. The verifier $R$ is algorithmically defined, ensuring verifiable reward computation. Both $\mathcal{P}$ and $R$ are implemented as programs. Figure 1(b) illustrates an example of a verifiable environment that asks the LM to sort a given array in ascending order. We also define an integer difficulty level $d \in [0, +\infty)$ for each environment to control the expected reasoning complexity for solving the generated problems. For example, in the sorting environment, a larger $d$ results in a longer array, as sorting a longer array typically requires stronger long-horizon reasoning.

Formally, a concrete problem is specified by parameters $p$ with environment-specific components, e.g., the length and elements of the array to be sorted in a specific problem. The parameters $p$ are randomly generated by the problem generator $\mathcal{P}_d$ conditioned on a specific difficulty level $d$, written as $p \sim \mathcal{P}_d$ as the program $\mathcal{P}_d$ defines an implicit parameter distribution. How $d$ influences the parameters $p$ sampled from $\mathcal{P}_d$ is specific to each environment. We denote this specific problem by $E_p = (I_p, R_p)$, where $I_p$ is the instantiated input obtained by filling the template with $p$, and $R_p$ is the corresponding verifier that computes a scalar reward $R_p(o) \in \mathbb{R}$ for an output $o$ to this specific problem.

The verifier extends the range of rewards adopted in prior works on RLVR (Lambert et al., 2025; DeepSeek-AI, 2025; Kimi Team, 2025b), as each environment independently defines its own verifier. For example, in an environment that asks the LM to solve a Sudoku puzzle, the verifier checks that the output satisfies all Sudoku constraints, rather than comparing it against a pre-computed correct solution, of which multiple, and sometimes many, may exist.

## 2.2. RL with Adaptive Verifiable Environments

When training a policy model $\pi$, RLVE maintains a difficulty range $[\ell_\pi, h_\pi]$ that governs problem generation within a specific verifiable environment, and dynamically adjusts this range based on the current performance of $\pi$. When $\pi$ performs well at the difficulty level $h_\pi$, $h_\pi$ is incremented to shift the difficulty distribution, as shown in Figure 2.

Specifically, we initially set $\ell_\pi = h_\pi = 0$, so training begins with the simplest problems available in the environment. When generating a new problem, a difficulty level $d$ is uniformly sampled from $[\ell_\pi, h_\pi]$, and the environment's problem generator samples parameters $p \sim \mathcal{P}_d$ to instantiate a concrete problem. We track two quantities: the number of correct rollouts $a$, and the total number of attempted rollouts $b$ across all problems sampled from $\mathcal{P}_{h_\pi}$. Whenever $b$ exceeds a minimum sample threshold $\tau_{\mathrm{num}}$, RLVE compares the observed accuracy $a/b$ against a predefined performance threshold $\tau_{\mathrm{acc}}$. If $a/b \geq \tau_{\mathrm{acc}}$, the model $\pi$ is considered proficient at this difficulty level, and the upper bound is incremented by one, i.e., $h_\pi \leftarrow h_\pi + 1$, thereby introducing more challenging problems. After this check, the statistics $(a, b)$ are reset, and the process continues.

RLVE does not impose a predefined cap on the upper bound $h_\pi$: within the available compute budget, $h_\pi$ naturally increases as long as $\pi$ continues to satisfy the performance criterion at successively higher difficulty levels. To prevent unbounded expansion of the difficulty range, which would reduce exposure to harder problems, RLVE uses a sliding window of difficulty levels by capping $\ell_\pi$ with a hyperparameter $d_\Delta > 1$: after each update of the upper bound $h_\pi$, we set the lower bound to $\ell_\pi = h_\pi - d_\Delta + 1$ whenever the range $h_\pi - l_\pi + 1$ exceeds $d_\Delta$. Intuitively, RLVE exposes $\pi$ to problems that are neither too easy nor too hard, since it has performed well on $\mathcal{P}_{h_\pi - 1}$ but not yet on $\mathcal{P}_{h_\pi}$.

RLVE naturally extends training from a single verifiable environment to multiple environments jointly (Algorithm 1). Specifically, given a collection of $n$ verifiable environments $\{E^{(1)}, E^{(2)}, \ldots, E^{(n)}\}$, RLVE first draws an environment $E^{(i)}$ uniformly from this collection when generating a new training problem; RLVE then follows the identical algorithm described above for the selected environment $E^{(i)}$. For each adaptive environment, RLVE maintains an independent difficulty range $[\ell_\pi^{(i)}, h_\pi^{(i)}]$, along with the statistics $(a^{(i)}, b^{(i)})$ for monitoring the model's performance at its current upper bound difficulty level $h_\pi^{(i)}$. In Section 3, we describe how the verifiable environments are constructed.

## 2.3. RL Algorithm

Any algorithm applicable to RLVR can be directly applied to RLVE. We adopt the DAPO algorithm (Yu et al., 2025), which is a variant of the GRPO algorithm (Shao et al., 2024).

*Table 1.* Six representative environment sources in RLVE-GYM, with one example environment per source.

| ENVIRONMENT SOURCE | DESCRIPTION OF EXAMPLE ENVIRONMENT |
|---|---|
| PROGRAMMING COMPETITION | `Count the permutations p of 1..{N} whose bubble-sort swap count equals a given lower bound, and that are lexicographically greater than {given_permutation}.` |
| MATHEMATICAL OPERATION | `Find an antiderivative F(x) such that F'(x) = {f_prime}.` |
| OPTIMIZATION PROBLEM | `Given f(x) = {polynomial}, find x0 that minimizes f(x).` |
| CLASSICAL ALGORITHMIC PROBLEM | `Sort the {N} numbers in ascending order: {array}.` |
| LOGICAL PUZZLE | `Solve the {NM}x{NM} Sudoku so every row, column, and each {N}x{M} subgrid contains 1..{NM}. Grid: {sudoku_puzzle}.` |
| NP-COMPLETE PROBLEM | `Given a directed graph with {N} vertices and edges {edges}, find a Hamiltonian path visiting every vertex exactly once.` |

We use the standard practice of DAPO's dynamic sampling: at each rollout step, we oversample rollouts using a prompt batch size larger than the training batch size and discard prompts with identical rollout rewards across all outputs; this process repeats until the training batch is fully populated. Additional details are provided in Appendix C.

In this context, we define the *effective prompt ratio* as the percentage of prompts whose rollouts yield non-identical rewards (and are therefore not discarded by DAPO's dynamic sampling). A high effective prompt ratio indicates that a large portion of problems are appropriately challenging for the policy. A lower ratio increases the time per training step, as dynamic sampling requires sending more prompts to the inference engine to obtain a single one that contributes to parameter updates; a higher ratio improves learning efficiency by reducing wasted rollouts from the inference engine, which typically constitutes the computational bottleneck in LM RL training (Hu et al., 2024).

## 3. RLVE-GYM: A Suite of 400 Environments Created through Environment Engineering

To implement RLVE, we carefully construct a suite of 400 verifiable environments, named RLVE-GYM. We give some representative sources of environments from RLVE-GYM and corresponding example environments in Table 1. We next introduce the principles and insights used in our **environment engineering**, with more details in Appendix B.

### 3.1. Environments as Pedagogical Tools

The first principle we used in environment engineering is to build pedagogical tools for developing reasoning capabilities. For example, if our goal were purely to obtain a sorted array, we would execute a sorting program, which is reliable and far more efficient than relying on the model's error-prone reasoning. Instead, we train the model to learn the reasoning process that underlies array sorting, rather than aiming to replace the sorting program with the model.

As a concrete example, many environments in RLVE-GYM

are adapted from programming competition problems. In the original problem setting, competitors are asked to write programs that correctly solve the provided test cases. Thus, if the goal were merely to obtain the correct output, one could simply execute such a program on the test case. Instead, the model is required to manually produce the correct outputs without executing code. Through this process, the model could learn reasoning capabilities that can generalize to broader tasks (Bao et al., 2025). For example, even manually simulating a simple recursive function for brute force could involve problem decomposition, self-verification, and backtracking (Gandhi et al., 2025), not to mention that successful manual solving often requires more sophisticated reasoning to complete a problem within a reasonable time.

### 3.2. Verification via Environment Advantages

The second principle we used in environment engineering is to verify model outputs by exploiting the advantages of the environment over the LM. In environments constructed under the first principle, writing problem-solving programs is usually easier and more reliable than solving problems manually. As the environment can execute programs while the LM is not allowed to do so, we can exploit this advantage by using such programs (e.g., those originally designed to solve programming problems) for output verification.

Another advantage is that the environment is responsible only for verifying outputs, whereas the LM is tasked with solving the problems. We exploit this advantage when building some environments that exhibit inherent asymmetry between solving and verification in computational complexity, allowing us to eliminate the need for implementing time-consuming solvers altogether. For example, in a Sudoku environment, generation can be performed by masking cells in a randomly sampled complete solution to form a valid puzzle; verification is straightforward using Sudoku rules, whereas solving the puzzle itself requires intractable time complexity, which the environment does not need to implement. NP-complete problems such as SAT (Cook, 1971) or Hamiltonian path detection (Garey & Johnson, 1979; Karp, 1972) typically exhibit this asymmetry. Another example is

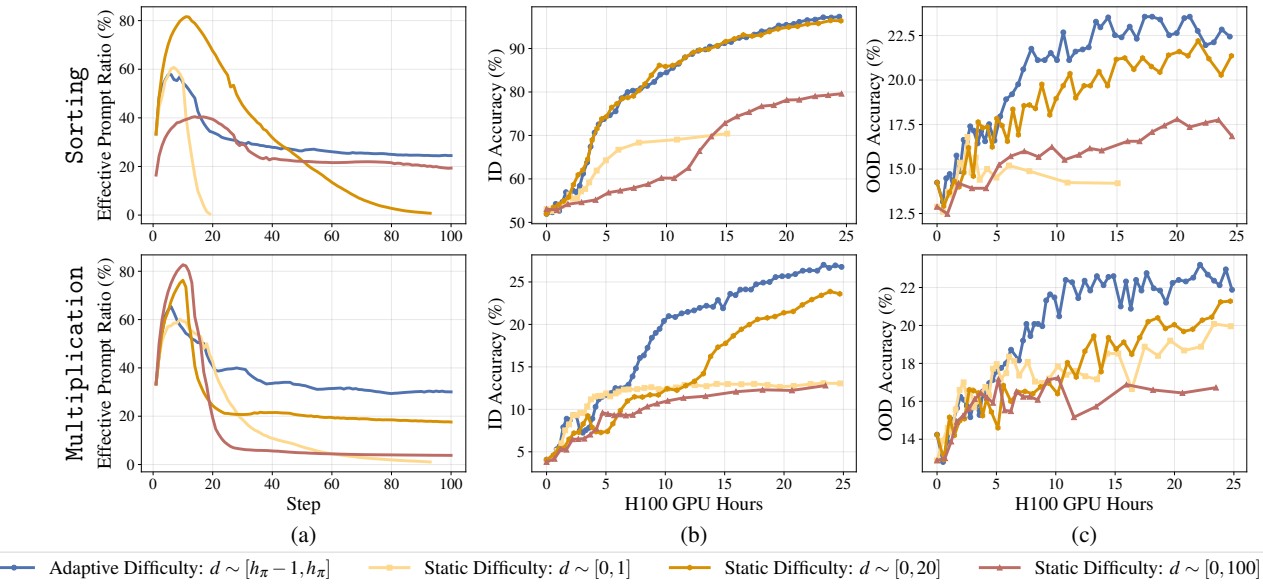

*Figure 3.* Comparison of RLVE (using dynamically adjusted difficulty range) against static difficulty ranges. (a) reports the effective prompt ratio, defined as the percentage of prompts retained after dynamic sampling whose rollouts yield non-identical rewards; a higher ratio indicates generally better learning efficiency. (b) shows in-distribution (ID) accuracies on the same training environment, and (c) shows out-of-distribution (OOD) accuracies on the 50 held-out environments. **Adaptive difficulty maintains the highest effective prompt ratio and achieves superior performance,** whereas static difficulty suffers from either early saturation or inefficient learning.

an environment where the model computes the integral of a function: the generator creates an elementary function $f$ and asks the model to compute the definite integral of $f'$; the verifier then simply checks whether the output corresponds to $f$, without computing the integral itself. Exploiting this advantage provides supervision signals that are otherwise infeasible to obtain through imitation learning, offering the long-term potential to train LMs on highly complex tasks that humans cannot easily solve themselves.

### 3.3. Designing Configurable Difficulty

We design how incrementing the difficulty level $d$ leads to harder sampled problems for each environment independently. We achieve this by ensuring that solving any lower-difficulty problem is reducible to, or a subproblem of, solving a higher-difficulty one from the same environment.

For example, in sorting, if a model can sort arrays of length $(N+1)$, it must also be able to sort any array of length $N$, since inserting the smallest element at the beginning of a length-$N$ array produces a length-$(N+1)$ array whose solution implies the former. Similarly, integrating all functions whose expression trees have $(N+1)$ nodes presupposes the ability to handle all functions with $N$ nodes (e.g., solving $\int(f+1)$ implies solving $\int f$). Thus, in these two environments, a larger difficulty level $d$ corresponds to a longer array length and a larger expression tree size, respectively.

## 4. Analyzing Components of RLVE

In Sections 4.1 and 4.2, we study (1) the comparison between adaptive and static environments and (2) the effect of scaling the collection of training environments, respectively.

To facilitate our study, we build our own test set as a controlled evaluation setup. Specifically, we randomly select 50 environments as held-out test environments from RLVE-GYM and sample 50 distinct problems per environment, resulting in a fixed test set $\mathcal{D}_{\text{ood}}$ of 2,500 problems in total. From the remaining 350 environments, we randomly select 256 environments to form a collection $\mathcal{C}_{256}$. In all following experiments, every training problem used by RLVE is generated from $\mathcal{C}_{256}$ or its subsets, depending on the specific experiment. This setup ensures that the constructed test set $\mathcal{D}_{\text{ood}}$ serves as an explicit out-of-distribution (OOD) evaluation that focuses on evaluating generalizable reasoning capabilities. Further details are provided in Appendix D.

We experiment with three types of LMs: (1) base model: `Qwen2.5-7B-Base` (Qwen Team, 2024); (2) SFT model: `R1-Distill-Qwen-1.5B` (DeepSeek-AI, 2025); (3) RL models: `DeepScaleR-1.5B` (Luo et al., 2025) and `ProRL-1.5B-v2` (Hu et al., 2025a).

### 4.1. Adaptivity for Unstalled and Efficient Learning

To examine the effect of adaptive difficulty enabled by RLVE, we compare training on the same verifiable environment with adaptive difficulty against training with static

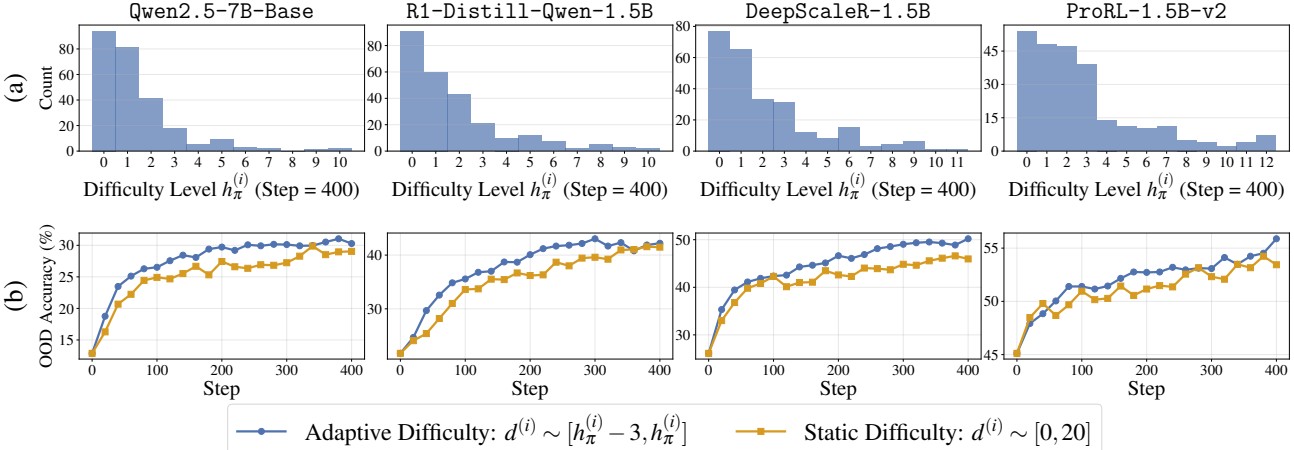

*Figure 4.* (a) shows the frequency distribution of the upper-bound difficulty levels $h_\pi^{(i)}$ reached by adaptive environments at step 400. (b) compares training jointly on 256 environments with adaptive versus static difficulty distributions. **Despite covering all adaptive environments' distributions, training on the static environments consistently underperforms.**

difficulty, where the difficulty distribution remains constant throughout training. We study three types of static difficulty distributions, where problem difficulty is uniformly sampled from $d \sim [0, 1]$, $d \sim [0, 20]$, and $d \sim [0, 100]$, respectively.

As a case study, we focus on training `Qwen2.5-7B-Base` (1) on `Sorting`, where the model is asked to sort a given array, and (2) on `Multiplication`, where the model is asked to compute the product of two integers, and higher difficulty corresponds to operands with more digits. For evaluation, we measure both in-distribution (ID) performance on a fixed set of held-out 4,000 problems generated from the same training environment with difficulty evenly sampled from $[0, 20)$, and OOD performance on $\mathcal{D}_{\text{ood}}$.

From Figure 3, we observe that when the static environment has a relatively low upper-bound difficulty, the effective prompt ratio eventually drops to zero, indicating that the model masters the hardest problems within the environment's static difficulty distribution after a certain amount of compute and subsequently becomes saturated. As a result, learning stalls once the environment ceases to provide learning signals. Prior works that train LMs on data generated by static verifiable environments have similarly been observed to suffer from early saturation (Li et al., 2025).

When the static environment instead has a high upper-bound difficulty such that the model cannot master all problems within a limited compute budget, the effective prompt ratio remains nonzero. However, the ratio drops substantially below that of adaptive difficulty, indicating only a small fraction of problems from the static environment are appropriately challenging. Empirically, this discrepancy significantly impairs both ID and OOD performance.

It is worth noting that training with the static difficulty distribution $d \sim [0, 20]$ confers an oracle advantage, as its dif-

ficulty distribution coincides with that of the ID evaluation. In realistic scenarios, finding such an oracle is infeasible. Even without such an oracle advantage, RLVE achieves comparable or even superior ID performance.

In conclusion, adaptive difficulty both (1) prevents learning from stalling due to overly easy problems and (2) avoids learning inefficiency caused by a large proportion of problems that are inappropriately challenging for the policy.

One might argue that an optimal static difficulty distribution could be manually tuned for a given compute budget. However, this manual tuning becomes infeasible when training jointly across many verifiable environments. To illustrate this, we train `Qwen2.5-7B-Base`, `R1-Distill-Qwen-1.5B`, `DeepScaleR-1.5B`, and `ProRL-1.5B-v2` using RLVE jointly across $\mathcal{C}_{256}$. Figure 4(a) shows the distribution of upper-bound difficulty levels reached by adaptive environments at step 400, revealing a broad range from 0 to 12. Using this distribution as oracle information, which would not even be available beforehand without adaptive difficulty, we train the same models on all environments from $\mathcal{C}_{256}$ using a static difficulty range of $[0, 20]$. As shown in Figure 4(b), although this static range covers every adaptive environment's difficulty distribution, training with such static environments is consistently outperformed by RLVE. These results suggest that the difficulty of each environment must be individually tuned to match the policy, as each environment defines its own notion of problem difficulty level; such a tuning arises naturally with adaptive environments enabled by RLVE.

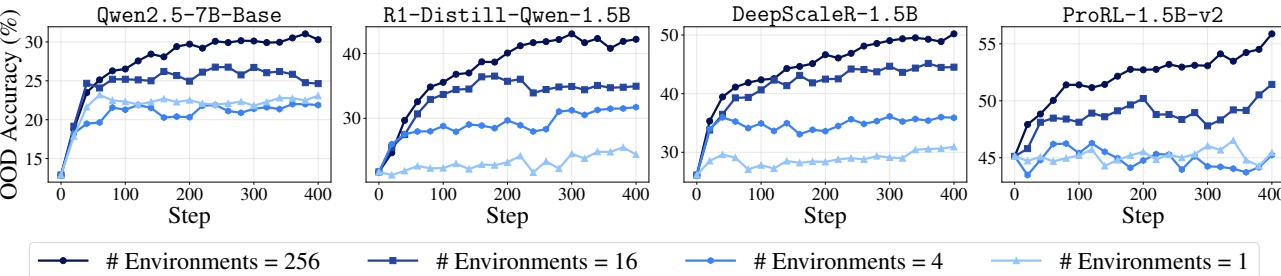

*Figure 5.* Comparison of RLVE with joint training on collections of four different sizes of verifiable environments, all under identical training setups. Each larger collection strictly contains all the smaller ones. Shown are the accuracies on 50 held-out environments. **Expanding the collection of training environments consistently improves performance on environments unseen during training.**

## 4.2. Environment Scaling as a Key Driver of Generalizable Reasoning Capabilities

We investigate the effect of scaling up the collection of verifiable environments used during training. Within $\mathcal{C}_{256}$, we construct three fixed environment collections: $\mathcal{C}_1$, $\mathcal{C}_4$, and $\mathcal{C}_{16}$, containing 1, 4, and 16 distinct verifiable environments, respectively; each larger collection contains all smaller ones, i.e., $\mathcal{C}_1 \subset \mathcal{C}_4 \subset \mathcal{C}_{16} \subset \mathcal{C}_{256}$. We train the same four models as in Section 4.1 separately on each environment collection under an identical setup, and evaluate checkpoints on $\mathcal{D}_{ood}$, which is constructed from the 50 held-out environments. Additional details are provided in Appendix E.

As shown in Figure 5, expanding the collection of training environments consistently leads to better performance on held-out environments across all model types. From another perspective, merely increasing the volume of RL training data remains insufficient for improving generalizable reasoning capabilities, given that a single environment can already generate an unbounded amount of data. Instead, scaling along the environment dimension emerges as a critical direction for future LM RL training. This insight echoes findings from classical RL research (Cobbe et al., 2020), and also resonates with observations from other LM training stages, such as SFT (Wang et al., 2022) and embedding learning (Su et al., 2023), where expanding the collection of tasks matters more than increasing the sheer volume of data.

## 5. Scaling Up RL Training with RLVE

After analyzing components of RLVE, we evaluate RLVE with joint training on all 400 verifiable environments from RLVE-GYM in two representative scenarios for scaling up RL training of LMs: (1) a data-saturation scenario, where the model has already saturated on a strong RLVR dataset (Section 5.1); and (2) a compute-constrained scenario, where training starts from a model without prior reasoning RL and the choice of RL data is crucial (Section 5.2).

In experiments for these two scenarios, we evaluate models on six representative reasoning benchmarks covering math-

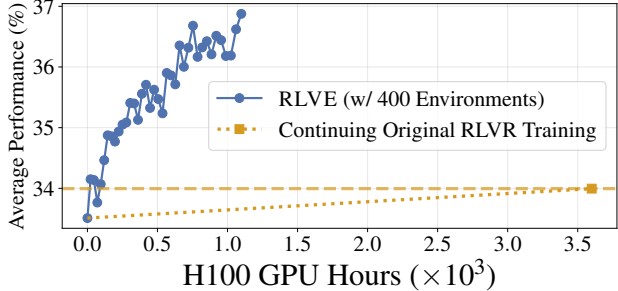

*Figure 6.* Comparison of RL training using RLVE jointly on all 400 verifiable environments in RLVE-GYM versus continuing original RLVR training. They both start from `ProRL-1.5B-v2` (Hu et al., 2025a), which was originally trained to saturation with RLVR on the ProRL dataset (Liu et al., 2025e). The checkpoint for continued original training, provided by Hu et al. (2025b), was obtained by further training on the same ProRL dataset. Shown is the average performance across six reasoning benchmarks throughout training. **RLVE effectively scales RL training when the model has already saturated on a strong RLVR dataset.**

ematics (AIME 2024/2025 (Art of Problem Solving, 2025), OMEGA-500 (Sun et al., 2025), and OlympiadBench (He et al., 2024)), code generation (LiveCodeBench (Jain et al., 2025)), and logical reasoning (BBEH (Kazemi et al., 2025)). Further evaluation details are provided in Appendix D.

### 5.1. Scaling beyond Data Saturation

We first evaluate to what extent RLVE enables further improvement beyond a model that represents one of the best open-sourced efforts to train an LM with RLVR at the time of our study. Specifically, we start from the checkpoint `ProRL-1.5B-v2` (Hu et al., 2025a). This starting checkpoint was originally trained from `R1-Distill-Qwen-1.5B` (DeepSeek-AI, 2025) using over 20,000 H100 GPU hours of RLVR, reaching performance saturation (Hu et al., 2025b) on its large and diverse training dataset of approximately 136,000 problems spanning mathematics, coding, logical reasoning, STEM, and instruction-following domains (Liu et al., 2025e); we refer to this dataset as the ProRL dataset for simplicity.

As a comparison, we also evaluate the checkpoint obtained by continuing the original RLVR training on the ProRL dataset, also initialized from `ProRL-1.5B-v2`. As shown in Figures 6 and 8, within approximately 1,100 H100 GPU hours, RLVE improves the average performance across the six reasoning benchmarks by an absolute 3.37%; in contrast, continuing the original RL training on the ProRL dataset yields only a marginal absolute improvement of 0.49% (7× smaller), even after using more than three times the compute (3,600 H100 GPU hours). Therefore, RLVE effectively scales up RL training beyond the data-saturation point of a model from one of the strongest open-sourced RLVR efforts to date. Importantly, the verifiable environments in RLVE-GYM are designed as pedagogical tools for developing generalizable reasoning capabilities rather than to resemble real LM tasks, and the resulting performance demonstrates effective transfer to real-world reasoning benchmarks.

## 5.2. Compute-Efficient Scaling

We next evaluate to what extent RLVE improves model performance in a compute-constrained scenario, where the available compute budget is fixed and the choice of RL data plays a crucial role. Specifically, we start from `OpenThinker3-1.5B` (Guha et al., 2025), which is currently the strongest open-sourced SFT model at the 1.5B scale and has not undergone any reasoning RL training. For comparison, we also train the LM on DeepMath-103K (He et al., 2025) under an identical training setup. DeepMath-103K is an RLVR dataset consisting of approximately 103K high-quality mathematical reasoning problems that are generally more challenging than those in comparable reasoning RLVR datasets, e.g., Luo et al. (2025); Yu et al. (2025).

As shown in Figures 7 and 9, training with RLVE consistently outperforms training on this existing high-quality RLVR dataset for the same number of steps. The model trained with RLVE consistently achieves higher performance on non-mathematical benchmarks (LiveCodeBench and BBEH) and most mathematical benchmarks (OMEGA-500 and OlympiadBench); RLVE also achieves comparable results on AIME 2024/2025, with peak performance exceeding DeepMath-103K by roughly one point. Thus, despite not targeting any specific benchmark domains, RLVE can significantly foster generalizable reasoning capabilities. Given that constructing DeepMath-103K required about $138,000 USD and 127,000 GPU hours (He et al., 2025), we offer a far more cost-efficient way to achieve stronger performance. Practically, these experiments also simulate the standard LM post-training pipeline, where LMs are first trained via SFT and subsequently via RL (DeepSeek-AI, 2025; Kimi Team, 2025a; Qwen Team, 2025; GLM-4.5 Team, 2025).

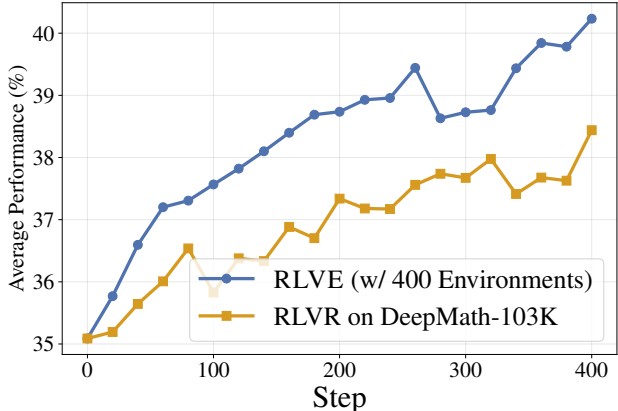

*Figure 7.* Comparison of RL training using RLVE jointly on all 400 verifiable environments in RLVE-GYM versus RLVR on the strong dataset DeepMath-103K (He et al., 2025), both starting from `OpenThinker3-1.5B` (Guha et al., 2025), the strongest open-sourced SFT model at the 1.5B scale. Shown is the average performance across six reasoning benchmarks throughout training. **Under an identical training setup, training with RLVE consistently outperforms the existing high-quality RLVR dataset.**

## 6. Related Work

**RL training of LMs on procedurally generated data.** Prior works have trained game agents (Cobbe et al., 2020) and, more recently, LMs (Hu et al., 2025c; Chen et al., 2025a; Liu et al., 2025d; Stojanovski et al., 2025; Li et al., 2025) via RL on procedurally generated data. However, they employ verifiable environments with static problem difficulty distributions, which, as shown by us, either suffer from early saturation (as in Li et al. (2025)) or inefficient learning. Among these works, Li et al. (2025) also observed that adding more environments improves performance, but their evaluations do not guarantee that test problems come from environments unseen during training; we show that such improvements generalize to entirely unseen environments. Liu et al. (2025c) trains LMs to solve SAT problems (Cook, 1971) with adaptive difficulty evolving with model performance. However, we have shown that only one environment is insufficient for developing generalizable reasoning capabilities. We therefore construct RLVE-GYM to scale up the collection of training environments.

**Adaptive data for LM training.** Curriculum learning has long been applied to improve RL efficiency by adjusting the data difficulty in increasing order throughout training (Baker et al., 2020; Wang et al., 2019; Portelas et al., 2020; Jiang et al., 2020; Gao et al., 2025), and has recently been extended to LMs (Kimi Team, 2025b; Shi et al., 2025; Chen et al., 2025b). These approaches operate on a finite dataset by reordering existing problems post hoc, whereas RLVE predefines difficulty levels over an infinite problem set from each verifiable environment and progresses successively through them. Cui et al. (2025); Yu et al. (2025)

dynamically filter out prompts that do not contribute to parameter updates. These approaches operate based on the results obtained after the rollout, complementing RLVE, which adapts the problems before being sent to the inference engine. It has also been explored that a trained and increasingly stronger LM makes the environment increasingly challenging in a self-play manner (Zhao et al., 2025; Liu et al., 2025a;b). In contrast, RLVE grounds environment adaptation in controllable manual construction, avoiding potentially incorrect LM-generated problems or verifiers. Besides RL training data, some work also tailors SFT data to model capability (Khan et al., 2025; Zeng et al., 2025).

## 7. Discussion on Future Work

**Scaling up RL with adaptive non-verifiable environments.** While this work focuses on verifiable environments, an equally important direction is to explore adaptive non-verifiable environments, such as creative writing or deep research (OpenAI, 2025; Shao et al., 2025), where rewards cannot be algorithmically defined. Non-verifiable environments tend to lack clear structure, which complicates difficulty control and constitutes an open research direction. Future work should also develop systematic principles for engineering non-verifiable environments. We believe that environment engineering will become as foundational to LM development as feature, data, and prompt engineering.

**Model-based automatic environment engineering.** In our preliminary explorations, we tried employing frontier LMs to perform automatic environment engineering. However, we found that maintaining high environment quality without human intervention proved nontrivial, particularly in ensuring (1) the unambiguity of the input template, (2) the reliability and efficiency of the problem generator in producing valid and diverse problems, and (3) the robustness of verifiers against diverse model outputs. For example, LMs often struggle to design environments that exploit the asymmetry between solving and verification in computational complexity, as such designs typically requires expert knowledge and deliberate engineering. We therefore consider the substantial human effort our authors devoted to manually engineering all 400 environments to be worthwhile. RLVE-GYM can serve as a prototype for future research on automatic environment engineering, analogous to how prior studies employed LMs for automatic data generation starting from a seed dataset (Wang et al., 2023).

## Acknowledgements

We thank Xiaoyu Chen, Zirui Cheng, Nouha Dziri, Scott Geng, Victoria Graf, Ronan Le Bras, Rulin Shao, Yijia Shao, Yizhong Wang, Teng Xiao, and Rui Xin for the helpful discussions. We thank Jiajun Li, Yuzhen Zhou, and Zilin Zhu for their help with implementing our experiments using the `slime` framework. We also thank Shizhe Diao and Jian Hu for providing the checkpoint obtained by continuing the original RLVR training of `ProRL-1.5B-v2` on the ProRL dataset. ZZ and YW are supported by Amazon AI Ph.D. Fellowships. SSL is supported by the Meta AI Mentorship program. JH is supported by an NSF Graduate Research Fellowship and the Meta AI Mentorship program. This work is also supported by NSF Grant Nos. IIS2142739, IIS2044660, and CHE2505932; by the Defense Advanced Research Projects Agency's (DARPA) SciFy program (Agreement No. HR00112520300); by the Singapore National Research Foundation and the National AI Group in the Singapore Ministry of Digital Development and Information under the AI Visiting Professorship Programme (award number AIVP-2024-001); by the AI2050 program at Schmidt Sciences; by a Google ML and Systems Junior Faculty Award; by gift funding from Ai2; by an Amazon AICE award; by the UW–Amazon Science Gift Hub; and by the UW–Tsukuba Amazon NVIDIA Cross Pacific AI Initiative (XPAI).

## Impact Statement

This paper presents work whose goal is to advance the field of Machine Learning. There are many potential societal consequences of our work, none which we feel must be specifically highlighted here.

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

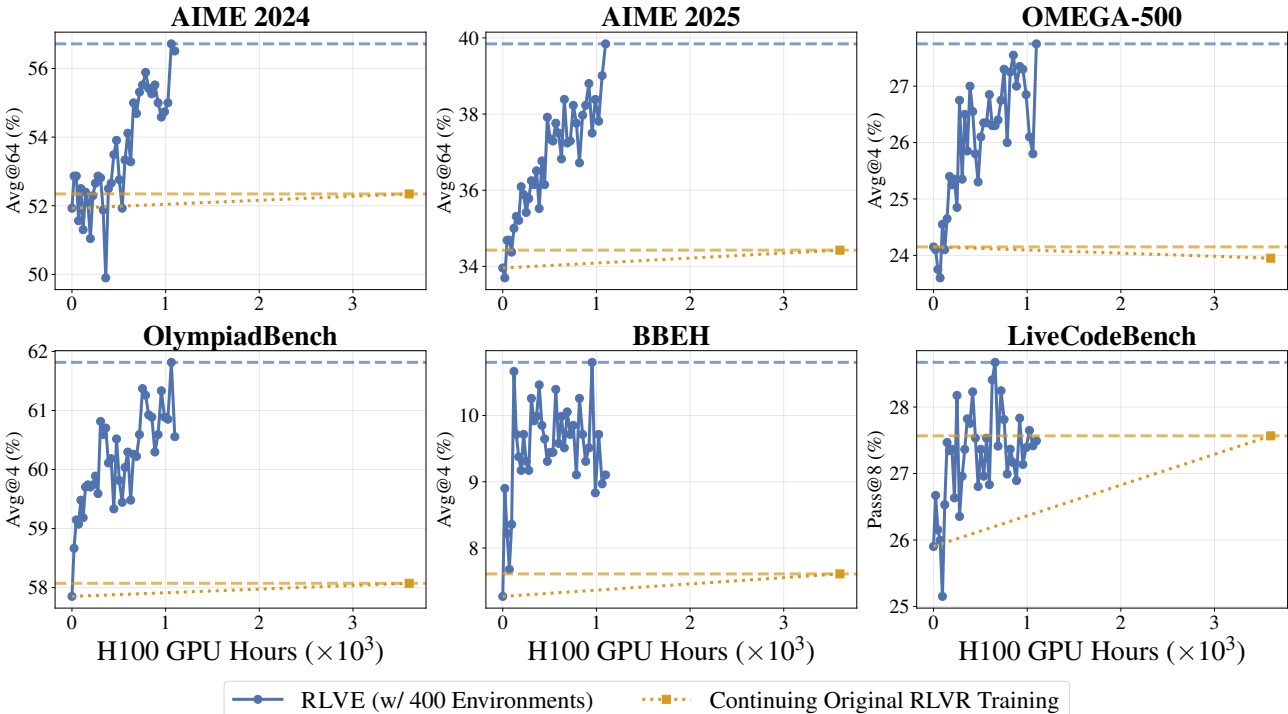

*Figure 8.* Results of Figure 6 shown separately for each of the six reasoning benchmarks, as detailed in Section 5. For clarity, each curve has a corresponding horizontal line indicating its highest point.

## A. Details of RLVE

The pseudocode for RLVE is provided in Algorithm 1. By default, we set the accuracy threshold $\tau_{\text{acc}}$ to 0.9, the minimum sample threshold $\tau_{\text{num}}$ to 8 times the number of rollouts per problem, and the sliding window size $d_\Delta$ to 4.

## B. Details of RLVE-GYM

### B.1. Representative Sources of Verifiable Environments with Example Environments in RLVE-GYM

In this subsection, we present six representative sources of verifiable environments from RLVE-GYM, providing one example for each source. Note that the reward range is always $[-1.0, +1.0]$ for each of the 400 verifiable environments.

**Verifiable environments inspired by programming competition problems.** A considerable portion of our verifiable environments is inspired by programming competition problems. We introduce the verifiable environment BubbleSwapLowerBound_PermutationCounting as an example, which is adapted from a programming problem in the Chinese National Olympiad in Informatics 2018 (CNOI 2018)[2]. The input template $I$ is:

```
Consider bubble sort on a permutation p[1..{N}] using the standard double loop:
```
for i = 1 to N:
  for j = 1 to N-1:
    if p[j] > p[j+1]: swap p[j], p[j+1]
```
It is known that the number of swaps performed by this algorithm is at least
LB(p) = (abs(1 - p[1]) + abs(2 - p[2]) + ... + abs(N - p[N])) / 2.
Tell me the number of permutations p of 1, 2, ..., {N} that satisfy BOTH:
1) The bubble sort swap count equals the lower bound: swaps(p) = LB(p).
2) p is lexicographically strictly greater than the given permutation P: {P}.
```

---

[2] https://noi.cn

---

**Algorithm 1** Pseudocode for RLVE (Reinforcement Learning with Adaptive Verifiable Environments). We focus on problem sampling, performance monitoring, and difficulty updates; any RL algorithm can be used to update the policy.

---

**Require:** policy model $\pi$, batch size $B$
**Require:** collection of $n$ verifiable environments $\{E^{(1)}, \ldots, E^{(n)}\}$ with $E^{(i)} = (I^{(i)}, \mathcal{P}^{(i)}, R^{(i)})$
**Require:** (hyperparameters) accuracy threshold $\tau_{\mathrm{acc}}$, minimum sample threshold $\tau_{\mathrm{num}}$, sliding window size $d_\Delta > 1$

1: **Initialize:** For each $i \in \{1, \ldots, n\}$, set $\ell_\pi^{(i)} \leftarrow 0$, $h_\pi^{(i)} \leftarrow 0$, $a^{(i)} \leftarrow 0$, $b^{(i)} \leftarrow 0$

2:

3: **while** training **do**

4:      $\mathcal{B} \leftarrow [\,]$ {Initialize empty batch (list)}

5:      **for** $j = 1$ to $B$ **do**

6:          Sample environment index $i \sim \mathrm{UniformInt}(\{1, \ldots, n\})$

7:          Sample difficulty $d \sim \mathrm{UniformInt}\big(\ell_\pi^{(i)}, h_\pi^{(i)}\big)$

8:          Generate problem: $p \sim \mathcal{P}_d^{(i)}$

9:          Instantiate: $I_p \leftarrow I_p^{(i)}, R_p \leftarrow R_p^{(i)}$

10:          Generate rollouts: $\mathcal{O} \leftarrow \mathrm{GenerateRollouts}(\pi, I_p)$

11:          Compute rewards: $\{r_o \leftarrow R_p(o) \mid o \in \mathcal{O}\}$

12:          Append to batch: $\mathcal{B}.\mathrm{append}\big((I_p, \mathcal{O}, \{r_o\})\big)$

13:          **if** $d = h_\pi^{(i)}$ **then**

14:             **for** each $o \in \mathcal{O}$ **do**

15:                $b^{(i)} \leftarrow b^{(i)} + 1$

16:                **if** $\mathrm{IsCorrect}(r_o)$ **then**

17:                    $a^{(i)} \leftarrow a^{(i)} + 1$

18:                **end if**

19:             **end for**

20:          **end if**

21:      **end for**

22:      Update policy: $\pi \leftarrow \mathrm{UpdatePolicy}(\pi, \mathcal{B})$

23:      **for** each environment $i = 1, \ldots, n$ **do**

24:          {Check and update difficulty}

25:          **if** $b^{(i)} \geq \tau_{\mathrm{num}}$ **then**

26:             **if** $a^{(i)}/b^{(i)} \geq \tau_{\mathrm{acc}}$ **then**

27:                $h_\pi^{(i)} \leftarrow h_\pi^{(i)} + 1$

28:                **if** $h_\pi^{(i)} - \ell_\pi^{(i)} + 1 > d_\Delta$ **then**

29:                    $\ell_\pi^{(i)} \leftarrow h_\pi^{(i)} - d_\Delta + 1$

30:                **end if**

31:             **end if**

32:             $a^{(i)} \leftarrow 0$, $b^{(i)} \leftarrow 0$

33:          **end if**

34:      **end for**

35: **end while**

36: **return** trained policy $\pi$

---

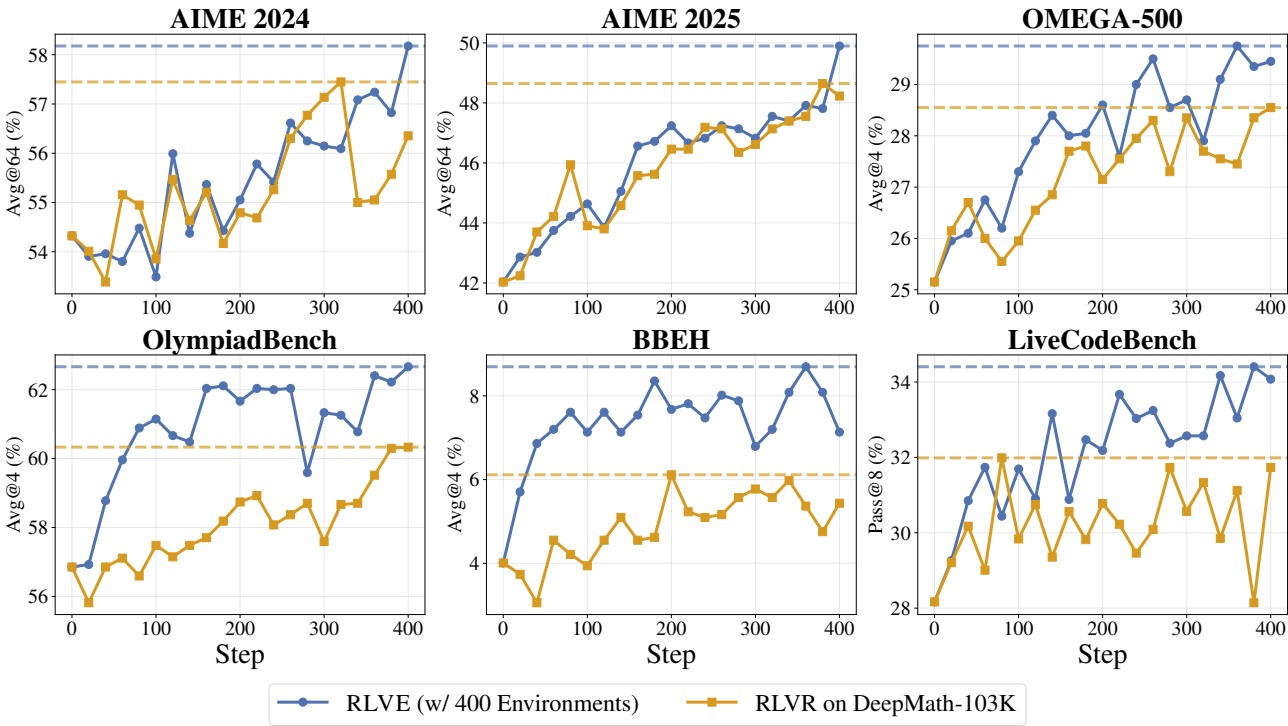

*Figure 9.* Results of Figure 7 shown separately for each of the six reasoning benchmarks, as detailed in Section 5. For clarity, each curve has a corresponding dotted horizontal line indicating its highest point.

In this environment, the problem specification parameters $p$ include the length $\{N\}$ and the given permutation $\{P\}$ used for the lexicographic constraint. The problem generator $\mathcal{P}_d$ sets the permutation length as $N = d + 3$, where $d \geq 0$ is the difficulty level, and uniformly samples a permutation $P$ of $\{1, 2, \ldots, N\}$. The verifier $R_p$ computes the correct answer using the algorithm that solves the original programming problem. Given a model output $o$, the verifier first attempts to extract a numeric answer from $o$. If the output format is invalid or the parsed answer is not a non-negative integer, the verifier assigns a reward of $-1.0$; otherwise, letting $x$ denote the correct answer and $y$ the model's predicted answer, the verifier computes the reward as $R_p(o) = (\min(x, y) / \max(x, y))^{10}$, which smoothly penalizes deviations from the correct answer.

**Verifiable environments of mathematical operations.** Some verifiable environments in RLVE-GYM focus on performing fundamental mathematical operations. As an illustrative example, we introduce the verifiable environment `Integral`, which asks the model to compute the indefinite integral of an elementary function. The input template $I$ is:

```
You are given the derivative of a function: F'(x) = {f_prime}

Your task is to find **an antiderivative** F(x) such that its derivative is equal to
the given expression.

**Output Format:** Your answer should be the expression for F(x), written in **SymPy
syntax**. Do not omit any symbols (e.g., always use '*' for multiplication).
Example: 'sin(2*x)/2' (do **NOT** include quotes or backticks).
```

In this environment, the problem generator $\mathcal{P}_d$ (conditioned on the difficulty level $d$) randomly generates an elementary function $F(x)$ by recursively constructing an expression tree whose node count equals $d + 2$. It then uses the SymPy[3] Python package to compute its derivative $F'(x)$, which is substituted into the input template to obtain the instantiated input $I_p$. The verifier $R_p$ parses the model's output $o$ into an expression and checks whether the derivative of the predicted function equals the provided $F'(x)$. If the output cannot be parsed into a valid expression, the verifier assigns a reward of $-1.0$; if the parsed expression correctly satisfies the condition, the reward is $+1.0$; otherwise, it is $0.0$. Note that this environment exploits the solving–verification asymmetry: we never need to compute the integral ourselves. If we were to

---

[3] https://www.sympy.org/en/index.html

instead randomly sample an elementary function $f(x)$ and attempt to compute its integral $\int f(x)\,dx$ directly, the resulting antiderivative might not admit a closed-form expression in terms of elementary functions; consequently, even symbolic solvers such as `SymPy` may be unable to produce an exact expression, whereas our verification is straightforward.

**Verifiable environments of optimization problems.** Some verifiable environments in RLVE-GYM are designed as optimization problems. We introduce the verifiable environment `PolynomialMinimum`. The input template $I$ is:

```
Given f(x) = {polynomial}, find the value of x0 that minimizes f(x).
Your final answer should be a single real number in decimal form, representing the
value of x0.
```

The problem generator $\mathcal{P}_d$ (conditioned on the difficulty level $d$) randomly generates a polynomial of degree $2(d+1)$, ensuring that the coefficient of the highest-order term $x^{2(d+1)}$ is positive so that $f(x)$ admits a global minimum. The verifier $R_p$ first attempts to extract a numeric value from the model's output $o$; if parsing fails, it assigns a reward of $-1.0$. Otherwise, let $x_0$ denote the model's predicted minimizer and $x_0'$ the true minimizer computed analytically or via a numerical solver. The verifier then evaluates both $f(x_0)$ and $f(x_0')$ and computes the reward as $R_p(o) = ((f(x_{\text{trivial}})-f(x_0))/(f(x_{\text{trivial}})-f(x_0')))^5$, where $x_{\text{trivial}}$ denotes a simple reference point (e.g., $x = 0$). This formulation smoothly rewards outputs that approach the true minimum, encouraging the model to identify increasingly accurate minima.

**Verifiable environments of classical algorithmic problems.** Some verifiable environments in RLVE-GYM are designed based on classical algorithmic problems. As an illustrative example, we introduce the verifiable environment `Sorting`, which asks the model to sort a given array of numbers in ascending order. The input template $I$ is:

```
You are given the following list of numbers:
{array}
Please sort them in **ascending order**.

**Output Format:**
Your final answer should be a single line containing the sorted numbers, separated by
spaces.
Example: 1 2 3 4 5
```

The problem generator $\mathcal{P}_d$ (conditioned on the difficulty level $d$) samples parameters $p$ such that the array length $N$ is roughly proportional to $3 \times 1.1^d$, and randomly generates the array elements. The verifier $R_p$ first checks the output format; if the model's output cannot be parsed into a valid list of numbers, it assigns a reward of $-1.0$. If the output array has an incorrect length, the reward is $-0.5$. Otherwise, let $x$ denote the number of positions where the predicted array elements match the array correctly sorted by a program. The verifier then computes the reward as $R_p(o) = (x/N)^{10}$.

**Verifiable environments of logical puzzles.** Some verifiable environments in RLVE-GYM are designed as logical puzzles. As an illustrative example, we introduce the verifiable environment `Sudoku`. The input template $I$ is:

```
Solve the following Sudoku puzzle of size ({N} x {M}) x ({M} x {N}) = {NM} x {NM}.
Each number is in the range from 1 to {NM}, and empty cells are represented by 0.
Here is the input grid:
{sudoku}

Rules of Sudoku:
1. Each **row** must contain all digits from 1 to {NM}, without repetition.
2. Each **column** must contain all digits from 1 to {NM}, without repetition.
3. The grid is divided into {M} x {N} **subgrids**, each of size {N} x {M}.
   Each subgrid must also contain all digits from 1 to {NM}, without repetition.

**Output Format:**
Your final answer should contain {NM} lines, each with {NM} numbers separated by
spaces.
The numbers should represent the completed Sudoku grid in **row-major order**,
matching the format of the given input (i.e., the first number on the first line
corresponds to the top-left cell of the grid).
```

The problem generator $\mathcal{P}_d$ (conditioned on the difficulty level $d$) samples parameters $p$ such that the larger of $N$ and $M$ does not exceed $d + 2$. It exploits the solving–verification asymmetry: instead of solving Sudoku puzzles from scratch, the generator first generates a complete valid Sudoku solution by applying a sequence of random equivalence transformations (e.g., row and column swaps within bands or stacks, symbol relabeling) to a canonical solved grid. It then masks a subset of cells randomly to form the partially filled puzzle. The verifier $R_p$ first checks the output format; if the model's output cannot be parsed into a valid grid of the expected dimensions, a reward of $-1.0$ is assigned. Otherwise, if the filled grid satisfies all Sudoku rules, the verifier returns a reward of $+1.0$; if any rule is violated, the reward is $0.0$. The implementations of some other puzzle environments reference existing works (Xie et al., 2024; Chen et al., 2025a).

**Verifiable environments of NP-complete problems.** Some verifiable environments in RLVE-GYM are designed based on NP-complete problems. Here, we present an example, `HamiltonianPathExistence`, that asks the model to find a Hamiltonian path in a given directed graph, which guarantees the existence of a Hamiltonian path. The input template $I$ is:

```
You are given a **directed graph** with {N} vertices labeled from `0` to `{N_minus_1}`.
The graph contains the following directed edges.
Each edge is represented as a tuple `(s, t)`, meaning there is a directed edge **from
vertex `s` to vertex `t`**:
{edges}

Please find a path `p_1, p_2, ..., p_{N}` such that the path **visits every vertex
exactly once** (revisiting vertices is NOT allowed).

**Output Format:**
Your final answer should be a single line containing the path in order: `p_1, p_2,
..., p_{N}`, separated by **spaces**.
Example: `0 2 1` (do **NOT** include backticks or quotes); this means the path starts
at vertex 0, then goes to vertex 2, and finally to vertex 1 (assuming 3 vertices in
total).
```

The problem generator $\mathcal{P}_d$ (conditioned on the difficulty level $d$) samples parameters $p$ such that the number of vertices is $N = d + 3$. It then exploits the solving–verification asymmetry: it first samples a random permutation of all vertices and adds directed edges between every pair of adjacent vertices in this permutation, thereby guaranteeing that the generated graph contains at least one Hamiltonian path; additional edges are then randomly added to the graph. The verifier $R_p$ first checks whether the model's output can be parsed into a valid list of integers; if not, it assigns a reward of $-1.0$. If the output sequence is not a valid permutation of all vertices, the reward is $-0.5$. Otherwise, let $x$ denote the number of consecutive vertex pairs in the predicted path whose corresponding directed edges actually exist in the graph; the verifier then computes the reward as $R_p(o) = (x/(N-1))^5$, which equals $1.0$ if and only if the path is a valid Hamiltonian path. We frequently exploit this solving–verification asymmetry when constructing verifiable environments of NP-complete problems, given that verification can be performed efficiently, while finding a valid solution remains intractable within current human knowledge.

### B.2. Full List of 400 Verifiable Environments from RLVE-GYM

We provide the complete list of all 400 verifiable environments included in RLVE-GYM in this subsection. Their detailed implementations are available in our public repository. We list the unique symbolic names of all 400 environments below.

(1) `ABProgramSimulation`, (2) `AddMultiple_Divisible_Counting`, (3) `AdditionTable`,
(4) `AlmostCompleteGraphCycleCounting`, (5) `AndOr_Sequence_Counting`,
(6) `AntiPalindromicSubstringCounting`, (7) `Axis_KCenter`, (8) `BAJBytecomputer`,
(9) `BEZMinimalistSecurity`, (10) `BannedPointSupersetPathCounting`, (11) `BanyanHeart`,
(12) `BezoutIdentity`, (13) `Binario`, (14) `Binario_NoAdjacencyRequirement`,
(15) `BinaryAlternation`, (16) `BinaryLinearEquation_SolutionCounting`,
(17) `BinaryTreeLeafNumExpectation`, (18) `BitAndZero_PathCounting`, (19) `BitEquationCounting`,
(20) `BitwiseOperationSequenceCounting`, (21) `BlockImage`,
(22) `BoundedAdjacencyDifference_Permutation_Counting`, (23) `BoundedIntervalIntersection`,
(24) `BoundedMeanSubarrayCounting`, (25) `BoundedSubarrayCounting`, (26) `BoxScheduling`,
(27) `Bridge`, (28) `BubbleSwapLowerBound_PermutationCounting`, (29) `BucketSorting`, (30) `CRT`,
(31) `CampfireParty`, (32) `CampsitePuzzle`, (33) `Canon`, (34) `CantorExpansion`,

(35) CapitalCityEffect, (36) CardColoringCounting, (37) CatalanNumberMod,
(38) CheckAllCycleXorZero, (39) ChoHamsters, (40) Cinema, (41) Circuit,
(42) CirculatingDecimalCounting, (43) CirculatingGrid, (44) CleaningUp, (45) ClearSymmetry,
(46) Clique_IndependentSet_Partitioning_Counting, (47) CoinSquareGame,
(48) ColoringCounting, (49) CombinationOddSubsequenceCounting,
(50) ConcatenationPartitionCountingSum, (51) CongruentEquation,
(52) ConstructHackInterval, (53) ConvexHull, (54) Cornfield, (55) CountdownClose,
(56) CountdownEqual, (57) CowDanceShow, (58) Cryptarithmetic,
(59) Cube_FixedLocalMaximumCounting, (60) CycleCounting, (61) DecreasingDigitCounting,
(62) DegreeFixed_SpanningTree, (63) DeltaMinPopcount, (64) DeltaNimGame,
(65) DerangementExtension, (66) DifferenceConstraintSystem,
(67) DifferenceConstraintSystemDAG, (68) DifferentColorPairing, (69) Differentiate,
(70) DigitLISCounting, (71) DiscreteLogarithm, (72) Disinfection,
(73) DistinctArrayPermutation, (74) DistinctEdgeColoredCompleteGraphCounting,
(75) Division, (76) DivisorFlipExpectation, (77) DoubleCrossCounting,
(78) DoublePalindromicStringCounting, (79) DoubleStackSorting, (80) DynDynamite,
(81) EightDigitPuzzle, (82) EmperorWorries, (83) EnergyStorageMeter, (84) EuclidGame,
(85) EvenDegreeGraphPartitioning, (86) Expression_AddingParenthese_Counting,
(87) FBI_BinaryTree, (88) FaceRightWay, (89) FactorialTrailingZeroCount, (90) Fibonacci,
(91) FibonacciContainingCounting, (92) Fibtrain, (93) FireworkShow,
(94) FixedModK_Selection_Counting, (95) FixedOneEdgeNum_SpanningTree,
(96) FractionalProgramming, (97) FractionalProgramming_BipartiteGraphMatching,
(98) FutoshikiPuzzle, (99) GCDFibonacciProduct, (100) GCDOne_Counting,
(101) GCDPrime_Counting, (102) GasFireExtinguishers, (103) GaussianElimination,
(104) GcdLcmCounting, (105) GoldWashing, (106) GraMinimaGame, (107) GradeRankingCounting,
(108) GraphContainTreeCounting, (109) GraphIsomorphism, (110) GridBFS,
(111) GridColoringCounting, (112) GridComponent, (113) GridLocalMinimumCounting,
(114) GridParityConstruction, (115) GridTriangleCounting, (116) HURWarehouseStore,
(117) HalvingChainCounting, (118) HamiltonianPath, (119) HamiltonianPathExistence,
(120) HeapCounting, (121) HitoriPuzzle, (122) HungryRabbit, (123) ImpParty,
(124) IndividualSumBounded_SequenceCounting, (125) IntegerFactorizationCounting,
(126) IntegerProgramming, (127) Integral, (128) InversionPair, (129) InversionPairK_Counting,
(130) Josephus, (131) JugPuzzle, (132) KPartition, (133) KUR, (134) Kakurasu, (135) KiddingMe,
(136) KingSorting, (137) KloBlocks, (138) Knapsack, (139) KnightsAndKnaves, (140) KosDicing,
(141) KthSubsequence, (142) Kth_BinaryTree, (143) Kth_SemiBalancedBracketSequence, (144) LAS,
(145) LASLaser, (146) LCM, (147) LDSTwo_Counting, (148) LIS_LDS_Concatenation, (149) LIZ_Lollipop,
(150) LampChanging, (151) LandAcquisition, (152) LandformGenerationCounting,
(153) LargestConvexPolygon, (154) LargestRectangle_AmongPoints, (155) LightUpPuzzle,
(156) LinkBeads, (157) LongestMaxDiffBoundedInterval, (158) LongestPath,
(159) Longest_DoublePalindrome, (160) Longest_MatchingSubsequence,
(161) Longest_RepeatedPalindrome, (162) MYJ, (163) MafMafia, (164) MagicSquarePuzzle,
(165) MakingGrade, (166) MatrixPermutationEquivalence,
(167) MatrixPermutation_BothDiagonalOne, (168) MatrixPermutation_MainDiagonalOne,
(169) MatrixPooling, (170) MatrixRMQCounting, (171) Matrix_BinaryExponentiation,
(172) MaxDifferentGroupPairDivision, (173) MaxGridPathIntersection,
(174) MaxMinimum_AfterIntervalAddition, (175) MaxMultSplit,
(176) MaxMultiplicationFixedSum, (177) MaxNoConflictingBombs, (178) MaxPermutation,
(179) MaxRMQExpectation, (180) MaxSegmentCoverageConstraint, (181) MaxSumLDS,
(182) MaxThreeSquareSum, (183) MaxTreeXorPath, (184) MaxTree_KPathCoverage,
(185) MaxWeightPalindromicSubstring, (186) MaxXorPath, (187) MaxXorSet,
(188) Max_NonAdjacent_KElementSum, (189) Max_TreeConstrainedPermutation_Weight,
(190) MaximumAchromaticNumber, (191) MaximumClique, (192) MaximumDivisor,
(193) MaximumIndependentSetGrid, (194) MaximumLexicographicalOrderSubsequence,
(195) MaximumPointSegmentMatching, (196) MaximumWeightMatching,

(197) Maximum_IndependentSet_Tree, (198) Maximum_SubsequenceNum, (199) Maze,
(200) MinConversionToCycleCost, (201) MinCostReducingLNDS, (202) MinCostTreeCoverage,
(203) MinCubeAssignment, (204) MinDivisionSumXor, (205) MinInorderBinaryTree,
(206) MinKDivisorNumber, (207) MinNoSolutionLinearDiophantineEquation,
(208) MinNonsubstring, (209) MinPairSumMultiplicationPermutation, (210) MinPathCover_DAG,
(211) MinSumChebyshevDistance, (212) MinSumDistanceSquare, (213) MinSumPreXor,
(214) MinSwapTwoPermutations, (215) MinXorPair, (216) Minesweeping, (217) MinimalCyclicShift,
(218) MinimumChromaticNumber, (219) MinimumChromaticNumber_SegmentOverlap,
(220) MinimumCost_MaximumFlow, (221) MinimumDirectedSpanningTree,
(222) MinimumFibonacciRepresentation, (223) MinimumHarmoniousChromaticNumber,
(224) MinimumIntervalCoverage, (225) MinimumRatioPath, (226) MinimumSpanningTree,
(227) MinimumSpanningTreeCounting, (228) MinimumSteinerTree,
(229) MinimumSumDifferenceSubmatrix, (230) MinimumTreeWeightedDominatingAncestor,
(231) MinimumUnconflictedGridKMax, (232) MinimumWeightedSpanningTree,
(233) Minimum_CrossingEdges_GraphPartition, (234) Minimum_DominatingInterval,
(235) Minimum_DominatingSet, (236) Minimum_DominatingSet_Grid, (237) Minimum_MaxAbsSlicer,
(238) Minimum_MaxSlicer, (239) Minimum_VertexCover, (240) MitterTransportation,
(241) MixedGraphEulerianCircuit, (242) MoneyChargingGame, (243) MonochromeBlockCounting,
(244) MonotonicStack, (245) MostComponentTreeRemovingTwoPaths,
(246) MostNumEdge_NonSelfIsomorphism, (247) MultiDrink, (248) MultipleFlippingGame,
(249) Multiplication, (250) NANDResultCounting, (251) NegativeBase, (252) NewNimGame,
(253) NextPalindromic, (254) NinePuzzle, (255) NoAdjacentGirlCounting,
(256) NoDoubleTripleCounting, (257) NotContainingStringCounting,
(258) NumberPartitionCounting, (259) Numbrix, (260) ODLDistance, (261) OddVisitation,
(262) PCPPermutation, (263) POLPolarization, (264) PairMoreOneCounting,
(265) PalembangBridges, (266) PalindromePartitionCounting,
(267) PalindromicSubstringNumberCounting, (268) PanSolarPanels,
(269) Path_NoGoingBack_Counting, (270) Patrol, (271) PipelineArrangement, (272) PolyaModel,
(273) PolynomialFactorization, (274) PolynomialInterpolation, (275) PolynomialMinimum,
(276) PolynomialRemainder, (277) PowerCycle, (278) PowerNest, (279) PowerShortcut,
(280) PrefixConcatenation, (281) PrefixProductMODDistinctPermutation,
(282) PrefixSumMODDistinctPermutation, (283) Prefixuffix, (284) PreorderTraversal,
(285) PrimeGraph_MinimumChromaticNumber, (286) ProtectingFlowers,
(287) PythagoreanGraph_IndependentSetCounting, (288) QuadMagicItems,
(289) QuadraticFunctionSegmentation, (290) QuantumLockPuzzle, (291) QueenPlacement,
(292) RandomRangeMaxExpectation, (293) RangeConstrained_IncreasingSequence_Counting,
(294) RangeFourSequenceConstruction, (295) RangeShrinkingSequenceCounting,
(296) RecursiveFunction, (297) RecursiveSequenceSumConstruction, (298) RepeatSequenceLNDS,
(299) RootExtraction, (300) RoundRobin, (301) RoundTableAssignment, (302) RoyalLockCounting,
(303) SAT, (304) SCC_Sequence_Counting, (305) SLOElephants, (306) STUWell, (307) SaladBar,
(308) SalesmanFatigue, (309) SameAdjacencyCounting, (310) SecretCowCode,
(311) SegmentMinLengthEqual_Counting, (312) SegmentTreeSortingCounting,
(313) SelfPowerSequenceMOD, (314) SetCover, (315) SetSplitting,
(316) SharedSubstringCounting, (317) ShortestPath, (318) ShortestPathCountConstruction,
(319) ShortestUnicolorSubstring, (320) SingingGirlStory, (321) SingleStackSorting,
(322) SkaRockGarden, (323) SkyscraperPuzzle, (324) SkyscraperSumPuzzle, (325) SlidingWindow,
(326) SmallestBinaryMultiple, (327) SmallestCircle, (328) Sorting, (329) SpiralMatrix,
(330) SplittingGame, (331) SpyNetwork, (332) SquSquarks, (333) SquareUndamagedPointCounting,
(334) StarBattle, (335) StirlingSecond, (336) StoneGame, (337) StoneIntervalsGame,
(338) StringPartitionShuffle, (339) StringReversalConstruction, (340) StuntFlying,
(341) SubarraySumXor, (342) SubarrayXorSum, (343) SubgraphIsomorphism,
(344) SubmatrixSumDivisibleCounting, (345) SubsequenceReversalLNDS, (346) SubsetSum,
(347) SubsetSumSequence, (348) Sudoku, (349) SumGCD, (350) SumGCDWithIndividual, (351) SumLCM,
(352) SumMOD, (353) SumManhattan_CurvedSurface, (354) SumPHIInterval,

(355) `SumProductDivisorNum`, (356) `SumPseudoEuclidean`, (357) `SumSetMultiplication`,
(358) `SumSpanningTreeGCD`, (359) `SumTriangleArea`, (360) `SumXorDivisorNum`, (361) `Sum_DivisorNum`,
(362) `SurvoPuzzle`, (363) `TakingPrimeGame`, (364) `TaskArrangement`, (365) `TetrisAttack`,
(366) `ThreeStringCommonSubsequenceCounting`, (367) `ThreeVertexCycleCounting`,
(368) `TopologicalSort`, (369) `TopologicalSort_MinimalLexicographicalOrder`,
(370) `Tournament_LongestPath`, (371) `TransmissionDelay`, (372) `TreeAddOneEdgeDiameter`,
(373) `TreeCenter`, (374) `TreeChangeOneEdgeDiameter`, (375) `TreeColoring`,
(376) `TreeDynamic_XORZeroPath`, (377) `TreeElimination_Expectation`,
(378) `TreeEvenPartitioning`, (379) `TreeMaximumVisitedVertex`,
(380) `TreeRandomWalkExpectation`, (381) `TreeTopologicalSequenceCounting`,
(382) `Tree_DistanceEqualTriad_Counting`, (383) `TriumphalArch`, (384) `TwiddlePuzzle`,
(385) `TwoSAT`, (386) `TwoSet_AllCoprime_Counting`, (387) `UndamagedSubmatrixCounting`,
(388) `ValueDiminishingSelection`, (389) `Vertex_KCenter`, (390) `VirusSynthesis`,
(391) `VisibleLine`, (392) `WIL`, (393) `WYC`, (394) `WYRLevelingGround`, (395) `WarehouseConstruction`,
(396) `WeightedBinaryTree`, (397) `WeightedLIS`, (398) `WhackAMole`, (399) `XorEquationCounting`,
(400) `ZeroPrefixSubsetCounting`.

## C. RL Training Details

We run our RL training using the `slime` framework[4] and adopt the DAPO algorithm (Yu et al., 2025), a variant of GRPO (Shao et al., 2024). Unless otherwise specified, the setup described below is consistent across all runs.

During each rollout step, we employ oversampling combined with dynamic filtering to enable dynamic sampling; we use a training batch size of 128 and an oversampling batch size of 384. We enable the partial-rollout technique (Kimi Team, 2025b; GLM-4.5 Team, 2025; Zhou et al., 2025), which caches unfinished generations from the current rollout step and subsequently resumes them. Each prompt produces 16 rollouts with a temperature of 1.0. We do not use KL regularization or entropy loss. The clipping range is $[0.2, 0.28]$. We adopt off-policy importance sampling (Yao et al., 2025) to correct for distribution mismatch between the training and inference engines. We perform one parameter update after each rollout step. We use the Adam optimizer (Kingma & Ba, 2015) with a constant learning rate schedule and weight decay of 0.01.

For `R1-Distill-Qwen-1.5B`, `DeepScaleR-1.5B`, `ProRL-1.5B-v2`, and `OpenThinker3-1.5B`, we use a learning rate of $2 \times 10^{-6}$ and a maximum rollout response length of 24,576 tokens. We use these models' default chat templates and directly insert the input into the user prompt field, following these models' prompt format for training.

For `Qwen2.5-7B-Base`, we use a learning rate of $1 \times 10^{-6}$ and a maximum rollout response length of 8,192 tokens. We adopt a variant of the prompt format from DeepSeek-AI (2025); Pan et al. (2025):

```
A conversation between User and Assistant. The user asks a question, and the assistant
solves it. The assistant first thinks about the reasoning process in the mind and then
provides the user with the answer. Show your work in <think> </think> tags, and return
the final answer in <answer> </answer> tags.
User: {input}
Assistant: Let me solve this step by step.
<think>
```

Every training run is conducted on a single node equipped with $8\times$ NVIDIA H100 (80GB) GPUs. The total training time of one single run varies depending on the number of training steps, the distribution of rollout response lengths, and the model size, ranging from approximately 2 to 8 days, which is equivalent to roughly 350 to 1,500 H100 GPU hours.

Details specific to RLVE are provided in Appendix A.

## D. Evaluation Details

We conduct all evaluations using the `SGLang` framework (Zheng et al., 2024) as the inference engine for model generation. We set the sampling temperature to $0.6$ and the top-$p$ parameter to $0.95$. For `R1-Distill-Qwen-1.5B`, `DeepScaleR-1.5B`, `ProRL-1.5B-v2`, and `OpenThinker3-1.5B`, the maximum response length during evaluation

---

[4]https://github.com/THUDM/slime

is 32,768 tokens; for `Qwen2.5-7B-Base`, the maximum response length is 16,384 tokens.

For AIME 2024/2025 (Art of Problem Solving, 2025), we sample 64 outputs per problem and report Avg@64 as the evaluation metric. For OMEGA-500 (Sun et al., 2025), OlympiadBench (He et al., 2024), and BBEH (Kazemi et al., 2025), we sample 4 outputs per problem and report Avg@4 as the evaluation metric. For LiveCodeBench (Jain et al., 2025), we sample 16 outputs per problem and report Pass@8 as the probability that at least one of 8 uniformly sampled outputs from the 16 outputs passes all test cases; we use the latest version (v6) of LiveCodeBench[5] available at the time of this work.

To construct the held-out test set $\mathcal{D}_{\text{ood}}$ introduced in Section 4, we randomly sample 50 held-out test environments from the full suite of 400 verifiable environments from RLVE-GYM (listed in Appendix B.2), which are:

(8) `BAJBytecomputer`, (10) `BannedPointSupersetPathCounting`, (11) `BanyanHeart`, (30) `CRT`,
(33) `Canon`, (42) `CirculatingDecimalCounting`,
(46) `Clique_IndependentSet_Partitioning_Counting`, (53) `ConvexHull`, (55) `CountdownClose`,
(57) `CowDanceShow`, (79) `DoubleStackSorting`, (80) `DynDynamite`,
(85) `EvenDegreeGraphPartitioning`, (86) `Expression_AddingParenthese_Counting`,
(89) `FactorialTrailingZeroCount`, (94) `FixedModK_Selection_Counting`, (98) `FutoshikiPuzzle`,
(106) `GraMinimaGame`, (115) `GridTriangleCounting`, (117) `HalvingChainCounting`, (123) `ImpParty`,
(125) `IntegerFactorizationCounting`, (131) `JugPuzzle`, (151) `LandAcquisition`,
(152) `LandformGenerationCounting`, (165) `MakingGrade`,
(167) `MatrixPermutation_BothDiagonalOne`, (171) `Matrix_BinaryExponentiation`,
(187) `MaxXorSet`, (197) `Maximum_IndependentSet_Tree`, (226) `MinimumSpanningTree`,
(234) `Minimum_DominatingInterval`, (245) `MostComponentTreeRemovingTwoPaths`,
(253) `NextPalindromic`, (266) `PalindromePartitionCounting`, (271) `PipelineArrangement`,
(274) `PolynomialInterpolation`, (280) `PrefixConcatenation`,
(287) `PythagoreanGraph_IndependentSetCounting`, (315) `SetSplitting`, (323) `SkyscraperPuzzle`,
(326) `SmallestBinaryMultiple`, (337) `StoneIntervalsGame`, (343) `SubgraphIsomorphism`,
(344) `SubmatrixSumDivisibleCounting`, (347) `SubsetSumSequence`, (355) `SumProductDivisorNum`,
(360) `SumXorDivisorNum`, (361) `Sum_DivisorNum`,
(369) `TopologicalSort_MinimalLexicographicalOrder`.

For each held-out test environment, we randomly generate 50 distinct problems, with the difficulty level $d$ evenly distributed within the range $[0, 4]$. This results in a total of 2,500 problems for the 50 held-out environments.

We sample one model output per problem for all evaluations on the in-distribution (ID) test set in 4.1 and on $\mathcal{D}_{\text{ood}}$.

# E. Details of Training Environment Collection

As described in Section 4, we construct four collections of training environments, denoted as $\mathcal{C}_1$, $\mathcal{C}_4$, $\mathcal{C}_{16}$, and $\mathcal{C}_{256}$, and each larger collection strictly contains all smaller ones, i.e., $\mathcal{C}_1 \subset \mathcal{C}_4 \subset \mathcal{C}_{16} \subset \mathcal{C}_{256}$. All four collections are from the full suite of 400 verifiable environments introduced in Appendix B.1, excluding the 50 held-out test environments.

For clarity, we list below the specific composition of each collection and its incremental difference from the preceding one.

$\mathcal{C}_1$ contains only a single environment (249) `Multiplication`.

$\mathcal{C}_4$ expands $\mathcal{C}_1$ by including 3 additional environments, for a total of 4. The incremental environments $\mathcal{C}_4 - \mathcal{C}_1$ are:

(75) `Division`, (84) `EuclidGame`, (328) `Sorting`.

$\mathcal{C}_{16}$ further expands $\mathcal{C}_4$ by adding 12 more environments, for a total of 16. The incremental environments $\mathcal{C}_{16} - \mathcal{C}_4$ are:

(100) `GCDOne_Counting`, (118) `HamiltonianPath`, (150) `LampChanging`, (153) `LargestConvexPolygon`,
(262) `PCPPermutation`, (269) `Path_NoGoingBack_Counting`, (303) `SAT`, (317) `ShortestPath`,
(329) `SpiralMatrix`, (345) `SubsequenceReversalLNDS`, (387) `UndamagedSubmatrixCounting`,
(394) `WYRLevelingGround`.

---

[5] https://huggingface.co/datasets/livecodebench/code_generation_lite/blob/main/test6.jsonl

$\mathcal{C}_{256}$ extends $\mathcal{C}_{16}$ by adding 240 additional environments, for a total of 256. The incremental environments $\mathcal{C}_{256} - \mathcal{C}_1$ are:

(1) `ABProgramSimulation`, (2) `AddMultiple_Divisible_Counting`, (3) `AdditionTable`,
(4) `AlmostCompleteGraphCycleCounting`, (6) `AntiPalindromicSubstringCounting`,
(9) `BEZMinimalistSecurity`, (13) `Binario`, (14) `Binario_NoAdjacencyRequirement`,
(15) `BinaryAlternation`, (16) `BinaryLinearEquation_SolutionCounting`,
(17) `BinaryTreeLeafNumExpectation`, (18) `BitAndZero_PathCounting`, (19) `BitEquationCounting`,
(20) `BitwiseOperationSequenceCounting`, (21) `BlockImage`, (23) `BoundedIntervalIntersection`,
(24) `BoundedMeanSubarrayCounting`, (25) `BoundedSubarrayCounting`, (26) `BoxScheduling`,
(27) `Bridge`, (31) `CampfireParty`, (35) `CapitalCityEffect`, (36) `CardColoringCounting`,
(38) `CheckAllCycleXorZero`, (40) `Cinema`, (41) `Circuit`, (43) `CirculatingGrid`, (44) `CleaningUp`,
(47) `CoinSquareGame`, (48) `ColoringCounting`, (50) `ConcatenationPartitionCountingSum`,
(51) `CongruentEquation`, (54) `Cornfield`, (56) `CountdownEqual`, (58) `Cryptarithmetic`,
(59) `Cube_FixedLocalMaximumCounting`, (60) `CycleCounting`, (61) `DecreasingDigitCounting`,
(62) `DegreeFixed_SpanningTree`, (63) `DeltaMinPopcount`, (64) `DeltaNimGame`,
(66) `DifferenceConstraintSystem`, (69) `Differentiate`, (71) `DiscreteLogarithm`,
(72) `Disinfection`, (76) `DivisorFlipExpectation`, (77) `DoubleCrossCounting`,
(78) `DoublePalindromicStringCounting`, (81) `EightDigitPuzzle`, (87) `FBI_BinaryTree`,
(88) `FaceRightWay`, (90) `Fibonacci`, (92) `Fibtrain`, (95) `FixedOneEdgeNum_SpanningTree`,
(96) `FractionalProgramming`, (97) `FractionalProgramming_BipartiteGraphMatching`,
(99) `GCDFibonacciProduct`, (102) `GasFireExtinguishers`, (104) `GcdLcmCounting`,
(107) `GradeRankingCounting`, (108) `GraphContainTreeCounting`, (109) `GraphIsomorphism`,
(110) `GridBFS`, (111) `GridColoringCounting`, (112) `GridComponent`,
(113) `GridLocalMinimumCounting`, (116) `HURWarehouseStore`, (119) `HamiltonianPathExistence`,
(121) `HitoriPuzzle`, (122) `HungryRabbit`, (124) `IndividualSumBounded_SequenceCounting`,
(126) `IntegerProgramming`, (127) `Integral`, (128) `InversionPair`, (129) `InversionPairK_Counting`,
(130) `Josephus`, (132) `KPartition`, (133) `KUR`, (136) `KingSorting`, (137) `KloBlocks`, (138) `Knapsack`,
(140) `KosDicing`, (141) `KthSubsequence`, (142) `Kth_BinaryTree`,
(143) `Kth_SemiBalancedBracketSequence`, (144) `LAS`, (145) `LASLaser`, (146) `LCM`,
(147) `LDSTwo_Counting`, (149) `LIZ_Lollipop`, (154) `LargestRectangle_AmongPoints`,
(155) `LightUpPuzzle`, (156) `LinkBeads`, (157) `LongestMaxDiffBoundedInterval`,
(159) `Longest_DoublePalindrome`, (160) `Longest_MatchingSubsequence`,
(161) `Longest_RepeatedPalindrome`, (164) `MagicSquarePuzzle`,
(166) `MatrixPermutationEquivalence`, (169) `MatrixPooling`, (170) `MatrixRMQCounting`,
(172) `MaxDifferentGroupPairDivision`, (173) `MaxGridPathIntersection`, (175) `MaxMultSplit`,
(178) `MaxPermutation`, (179) `MaxRMQExpectation`, (180) `MaxSegmentCoverageConstraint`,
(181) `MaxSumLDS`, (182) `MaxThreeSquareSum`, (185) `MaxWeightPalindromicSubstring`,
(186) `MaxXorPath`, (188) `Max_NonAdjacent_KElementSum`, (193) `MaximumIndependentSetGrid`,
(194) `MaximumLexicographicalOrderSubsequence`, (195) `MaximumPointSegmentMatching`,
(196) `MaximumWeightMatching`, (198) `Maximum_SubsequenceNum`, (200) `MinConversionToCycleCost`,
(201) `MinCostReducingLNDS`, (202) `MinCostTreeCoverage`, (203) `MinCubeAssignment`,
(204) `MinDivisionSumXor`, (206) `MinKDivisorNumber`,
(207) `MinNoSolutionLinearDiophantineEquation`, (208) `MinNonsubstring`,
(209) `MinPairSumMultiplicationPermutation`, (212) `MinSumDistanceSquare`,
(214) `MinSwapTwoPermutations`, (215) `MinXorPair`, (216) `Minesweeping`,
(218) `MinimumChromaticNumber`, (219) `MinimumChromaticNumber_SegmentOverlap`,
(221) `MinimumDirectedSpanningTree`, (222) `MinimumFibonacciRepresentation`,
(223) `MinimumHarmoniousChromaticNumber`, (225) `MinimumRatioPath`,
(227) `MinimumSpanningTreeCounting`, (229) `MinimumSumDifferenceSubmatrix`,
(230) `MinimumTreeWeightedDominatingAncestor`, (232) `MinimumWeightedSpanningTree`,
(233) `Minimum_CrossingEdges_GraphPartition`, (235) `Minimum_DominatingSet`,
(237) `Minimum_MaxAbsSlicer`, (238) `Minimum_MaxSlicer`, (240) `MitterTransportation`,
(241) `MixedGraphEulerianCircuit`, (243) `MonochromeBlockCounting`, (247) `MultiDrink`,
(248) `MultipleFlippingGame`, (251) `NegativeBase`, (255) `NoAdjacentGirlCounting`,

(256) `NoDoubleTripleCounting`, (258) `NumberPartitionCounting`, (259) `Numbrix`,
(260) `ODLDistance`, (263) `POLPolarization`, (264) `PairMoreOneCounting`, (265) `PalembangBridges`,
(267) `PalindromicSubstringNumberCounting`, (268) `PanSolarPanels`, (270) `Patrol`,
(273) `PolynomialFactorization`, (276) `PolynomialRemainder`, (277) `PowerCycle`, (278) `PowerNest`,
(279) `PowerShortcut`, (281) `PrefixProductMODDistinctPermutation`,
(282) `PrefixSumMODDistinctPermutation`, (283) `Prefixuffix`, (284) `PreorderTraversal`,
(285) `PrimeGraph_MinimumChromaticNumber`, (290) `QuantumLockPuzzle`, (291) `QueenPlacement`,
(292) `RandomRangeMaxExpectation`, (293) `RangeConstrained_IncreasingSequence_Counting`,
(294) `RangeFourSequenceConstruction`, (299) `RootExtraction`, (300) `RoundRobin`,
(301) `RoundTableAssignment`, (302) `RoyalLockCounting`, (304) `SCC_Sequence_Counting`,
(307) `SaladBar`, (308) `SalesmanFatigue`, (310) `SecretCowCode`,
(311) `SegmentMinLengthEqual_Counting`, (312) `SegmentTreeSortingCounting`,
(313) `SelfPowerSequenceMOD`, (316) `SharedSubstringCounting`,
(319) `ShortestUnicolorSubstring`, (321) `SingleStackSorting`, (324) `SkyscraperSumPuzzle`,
(325) `SlidingWindow`, (327) `SmallestCircle`, (330) `SplittingGame`, (331) `SpyNetwork`,
(332) `SquSquarks`, (334) `StarBattle`, (335) `StirlingSecond`, (336) `StoneGame`,
(338) `StringPartitionShuffle`, (346) `SubsetSum`, (348) `Sudoku`, (350) `SumGCDWithIndividual`,
(351) `SumLCM`, (352) `SumMOD`, (353) `SumManhattan_CurvedSurface`, (354) `SumPHIInterval`,
(356) `SumPseudoEuclidean`, (359) `SumTriangleArea`, (362) `SurvoPuzzle`, (363) `TakingPrimeGame`,
(364) `TaskArrangement`, (365) `TetrisAttack`, (367) `ThreeVertexCycleCounting`,
(370) `Tournament_LongestPath`, (371) `TransmissionDelay`, (372) `TreeAddOneEdgeDiameter`,
(373) `TreeCenter`, (374) `TreeChangeOneEdgeDiameter`, (375) `TreeColoring`,
(376) `TreeDynamic_XORZeroPath`, (377) `TreeElimination_Expectation`,
(378) `TreeEvenPartitioning`, (379) `TreeMaximumVisitedVertex`,
(380) `TreeRandomWalkExpectation`, (381) `TreeTopologicalSequenceCounting`,
(382) `Tree_DistanceEqualTriad_Counting`, (383) `TriumphalArch`, (384) `TwiddlePuzzle`,
(385) `TwoSAT`, (386) `TwoSet_AllCoprime_Counting`, (388) `ValueDiminishingSelection`,
(389) `Vertex_KCenter`, (391) `VisibleLine`, (392) `WIL`, (393) `WYC`, (396) `WeightedBinaryTree`,
(397) `WeightedLIS`, (399) `XorEquationCounting`, (400) `ZeroPrefixSubsetCounting`.

