# OpenReview forum: "RLVE: Scaling Up Reinforcement Learning for Language Models with Adaptive Verifiable Environments"
_ICML.cc/2026/Conference — ICML 2026 regular_

### Official Review · Reviewer_ryUU · 2026-02-27

**Soundness:** 3
**Presentation:** 3
**Significance:** 3
**Originality:** 4
**Overall Recommendation:** 4
**Confidence:** 4

**Summary:**

This work introduces RLVE, a framework that scales up reinforcement learning for language models by using adaptive verifiable environments to bypass the limitations of static datasets. By utilizing RLVE-GYM, a suite of 400 manually engineered environments covering tasks like programming, math, and logic, the system procedurally generates an infinite supply of problems with algorithmically verifiable rewards. Crucially, RLVE dynamically adjusts problem difficulty based on the model's evolving performance, ensuring it always trains at its "capability frontier" to maintain high learning efficiency and avoid saturation. Experimental results demonstrate that this approach consistently improves generalizable reasoning capabilities, allowing a 1.5B parameter model to achieve significant performance gains on real-world benchmarks while using substantially less compute than traditional methods.

**Compliance With Llm Reviewing Policy:**

Affirmed.

**Key Questions For Authors:**

(see weaknesses)

**Limitations:**

yes

**Strengths And Weaknesses:**

Strenghs:
1. It presents RLVE-GYM, a large-scale suite of 400 diverse, manually engineered environments that provide an infinite supply of algorithmically verifiable data, proving that scaling the number of environments is a key driver for generalizable reasoning.

Weaknesses:
1. What is the difference between allowing models to write code to solve the tasks versus restricting them from generating code? I have come across work with a similar setup (https://arxiv.org/pdf/2508.09125), where tasks are derived from code programs but LLMs are required to complete them purely through natural language reasoning rather than code generation, which is similar as yours. Do you have insights into what specific capabilities models may develop through such code-oriented tasks under this restriction? It would be interesting to see a discussion of this design choice and its implications and related work added to the paper.
2. Can models trained under this experimental setup generalize to other reasoning benchmarks? The benchmarks currently included in the paper appear closely aligned with the training data—primarily involving code and mathematical reasoning. It would be valuable to see additional results and analysis on more diverse reasoning benchmarks to better assess the robustness and transferability of the learned capabilities.

---

> ### Author Rebuttal · Authors · 2026-03-27
>
> We sincerely thank Reviewer ryUU for the insightful review. We appreciate the recognition of RLVE’s novelty and effectiveness. Below, we address the concerns in turn.
>
> + Reviewer ryUU asks about the difference between allowing models to write code versus requiring pure reasoning (i.e., language-only reasoning) only.
>   + We discuss this design choice in Section 3.1. Our main motivation is that restricting code execution, which deliberately makes the tasks more challenging, encourages the model to develop reasoning capabilities that are more generalizable and broadly transferable.
>     + For example, solving a sorting task programatically is trivial (e.g.,`sorted(array)`), whereas solving it manually requires substantially more complicated reasoning.
>     + A less trivial example is: `given two specific arrays, find the smallest set of indices to swap so that both arrays contain no duplicates.` This can be solved with a short Python program, but solving it purely by reasoning requires much more sophisticated capabilities, including decomposition, self-verification, and backtracking; here is an [example](https://ibb.co/TBX40cYT). In addition, for harder problems, efficient manual solving often also requires stronger long-horizon reasoning.
>     + We believe such capabilities are more generalizable and therefore more useful for real-world reasoning tasks. This is consistent with our results in Section 5, where RLVE improves performance on real-world reasoning-intensive benchmarks (AIME, LiveCodeBench, BBEH, etc).
>   + A second motivation is adaptive difficulty. Our goal is to build environments whose difficulty can be dynamically matched to the policy model.
>     + In the examples above, if code generation is allowed, increasing the input size often does not meaningfully increase solution difficulty, since the same short program still solves the problem.
>     + In contrast, when using pure reasoning, increasing input size directly increases reasoning difficulty. For example, manually sorting 10 numbers clearly requires more long-horizon reasoning than sorting 3, while the code solution remains essentially unchanged. This makes solving by pure reasoning much better suited for our setting of adaptive environments.
>   + We will clarify these points in the camera-ready version. We also thank the reviewer for pointing out the related work and will add it to the Related Work section. Specifically, both works adopt procedural generation, but our work focuses on training whereas theirs focuses on evaluation, and our work targets reasoning capability whereas theirs focuses on instruction-following capability.
>
> + Reviewer ryUU asks whether models trained under this setup generalize beyond math and coding.
>   + We agree that broader evaluation is important. In practice, however, mainstream LM reasoning benchmarks still primarily focus on math and coding because they are easy to verify, so our evaluation follows common practice in the literature.
>   + To partially mitigate this limitation, in the paper, we already conduct evaluations on BBEH, which focuses on logical reasoning.
>   + Following Reviewer ryUU’s suggestions, we also conducted an additional experiment on GPQA Diamond, which targets STEM reasoning. ProRL-1.5B-v2 obtains 41.4%, continued training on the original setup obtains 41.5%, and continued training with RLVE obtains 42.0%. The gain is smaller than on math, coding, and logical reasoning benchmarks. We believe this is because STEM reasoning also depends on substantial domain knowledge, whereas RLVE-GYM primarily trains general reasoning capabilities such as decomposition, self-verification, and backtracking, which are helpful but not sufficient on their own. We view extending RLVE toward broader knowledge-intensive reasoning as an important future direction.

---

> > ### Author Rebuttal · Reviewer_ryUU · 2026-04-01
> >
> > The authors’ responses regarding the restriction on code generation, comparisons to related work, and the additional experiments on GPQA-Diamond are reasonable and convincing. I would appreciate seeing these improvements incorporated into the next version of the paper. I will maintain my current score. Best of luck.

---

> > > ### Author Response · Authors · 2026-04-01
> > >
> > > Thanks very much! We are glad that all of your concerns have been addressed.

---

### Official Review · Reviewer_q5LJ · 2026-03-10

**Soundness:** 2
**Presentation:** 3
**Significance:** 2
**Originality:** 2
**Overall Recommendation:** 3
**Confidence:** 5

**Summary:**

The paper propose a set of configurable environments for RLVR. The objective is to conduct curriculum learning, where the environment will start from easy problems, and gradually increase the difficulty of the problems when the policy get better along the way. The paper introduces 400 environments through manual engineering.

**Compliance With Llm Reviewing Policy:**

Affirmed.

**Final Justification:**

I bump the score after rebuttal. But not sufficient for acceptance.

**Key Questions For Authors:**

NA

**Limitations:**

yes

**Strengths And Weaknesses:**

* Strength
- The paper a smooth curriculum learning that control difficulty through programmable configurations so that RL runs more efficiently, reaching new highs faster than standard RL approach.
- The paper is written well, easy to understand

* Weaknesses
- The methodology is not really novel. Curriculum learning and upward difficulty control has long been standard and explored by other RL-related works. And such strategy help improve efficiency.
- The environments are manually engineered and not scalable.
- The environments described are programmable, and not challenging in terms of semantics and linguistically reasoning heavy. There many tasks that cannot be configured or difficulty-controlled this way.
- No competitive curriculum learning or difficulty-control baselines are run and compared, such as simple difficulty or pass-rate filtering
- The experiments are done in 1B scale, not  showing stronger baselines.

---

> ### Author Rebuttal · Authors · 2026-03-27
>
> We sincerely thank Reviewer q5LJ for the constructive review with criticisms, which helps us improve the paper. Below, we address the concerns in turn.
> + Reviewer q5LJ questions the novelty of RLVE relative to existing curriculum learning strategies, which have long been used to improve training efficiency.
>   + **We discuss this distinction in the second paragraph of Appendix A**. We will move this discussion up to Related Work.
>   + Specifically, curriculum learning strategies typically operate on a finite dataset by reordering existing problems post hoc, whereas RLVE defines difficulty levels over an effectively infinite problem space induced by each verifiable environment and trains by progressively advancing through each level.
>   + Importantly, **RLVE is not only about improving training efficiency**, as in existing curriculum learning work. Each environment in our work can generate an effectively **infinite number of problems with unbounded difficulty. As a result, it is impractical to first materialize such problems as a finite dataset and then apply existing curriculum-learning methods**. RLVE enables keeping learning progressively rather than being capped by a fixed difficulty ceiling.
> + Reviewer q5LJ raises concerns about the cost, and thus scalability, of manual environment engineering.
>   + Our contribution is both the RLVE methodology and the RLVE-GYM environment suite. **RLVE itself is compatible with both manually or automatically constructed environments.**
>   + We also discuss this in Section 7. In brief, we did explore automatic environment construction with frontier models, but found it difficult to consistently maintain sufficiently high environment quality. We therefore consider the authors’ manual effort worthwhile as a strong prototype for future work on automatic environment generation; **rather than viewing the manually engineered environments merely as a limitation, we believe RLVE-GYM itself is an important contribution to the community**.
>   + We view manual engineering as a practical way at the present time, not as a necessary long-term requirement. Analogously, training data were once largely hand-curated, whereas synthetic data are now common. Given the rapid progress of frontier LMs in coding, we expect the scalability limitation of environment construction should not remain a major obstacle.
>   + In fact, even manually engineering RLVE-GYM does not incur an excessive cost. As a reference, building one environment took roughly 1 hour on average for a CS PhD student. As a rough upper-bound estimate, a CS Ph.D. student in the U.S. is typically paid less than 50USD per hour. The total human-labor cost of constructing RLVE-GYM is roughly bounded by 20,000USD, which is substantially smaller than the reported cost of DeepMath-103K.
> + Reviewer q5LJ raises the concern that the procedurally generated problems may be limited in semantic or linguistically heavy reasoning.
>   + While we agree that improving semantic and linguistic capabilities is also important, this is beyond the scope of this paper.
>   + We will make our scope clearer by further emphasizing both our motivation (scaling RLVR data for problem-solving reasoning) and our evaluation on real-world reasoning-intensive benchmarks.
> + Reviewer q5LJ notes that many tasks cannot be difficulty-controlled in our way.
>   + We agree and **discuss this as an important future direction in Section 7**, where we note that non-verifiable environments often lack clear structure, making difficulty control substantially harder and leaving this as an open research direction.
>   + That said, although the environments that we study cannot exhaustively cover all real-world tasks, RLVE-GYM helps models acquire reasoning capabilities that generalize beyond the training environments (Section 3.1), so RLVE significantly improves performance on reasoning-intensive real-world benchmarks (Section 5).
> + Reviewer q5LJ asks about comparisons with other curriculum-learning algorithms, such as pass-rate filtering.
>   + To clarify, **this work does not aim to propose a new curriculum-learning algorithm for arbitrary fixed datasets**. Rather, the goal is to study **scaling RL data via environments that generate an effectively infinite number of problems with unbounded difficulty**, with adaptive difficulty as a key component of the methodology.
>   + We also note that **we apply dynamic prompt filtering in all experiments, i.e., pass-rate filtering** that removes prompts that do not contribute useful gradient signals (Section 2.3), which has become a standard curriculum-learning practice in modern LM RL training. Therefore, in fact, **Section 4.1 already compares RLVE against a bounded-difficulty setup combined with an existing curriculum-learning strategy**. The empirical results show that RLVE is consistently better.
> + Reviewer q5LJ states that the experiments are only conducted at the 1B scale.
>   + Actually, the paper already includes experiments with Qwen2.5-7B-Base.

---

> > ### Author Rebuttal · Reviewer_q5LJ · 2026-04-02
> >
> > Thanks authors for the rebuttal. My concerns remain the same. In principle the "dfficulty" is artificially generated and increased over time. So initially we have tasks like "a+b?", then later we create tasks like "a+b+c+d+e+f......?" so it's more difficult. So it needs some analysis with curriculum learning

---

> > > ### Author Response · Authors · 2026-04-03
> > >
> > > We sincerely thank Reviewer q5LJ for the follow-up! However, we are still **unsure what concrete concerns Reviewer q5LJ continues to have, especially given that related concerns raised by other reviewers have been fully addressed**.
> > >
> > > Our paper already provides analyses that cover **both the necessity of dynamic environments and comparisons against existing curriculum learning strategies such as pass-rate filtering**:
> > >
> > > + In **Figure 1(a)**, we show that under our configurable difficulty design, some problems that are initially appropriately challenging become too easy during RL training, while others that were previously too hard become learnable as the policy improves. This supports both (1) that our environment difficulty design does track capability improvement, and (2) the need for environments that adapt dynamically to the policy model.
> > > + In **Figure 2**, we show that the model’s performance on problems at the current difficulty level gradually increases during RL training, and once it reaches the threshold, the environment increases the difficulty and performance correspondingly drops, after which the process repeats. This directly illustrates both (1) how the dynamic environment operates, and (2) how model performance evolves under this adaptive mechanism.
> > > + In **Section 4.1 (Figure 3)**, we compare RLVE against a bounded-difficulty setup combined with a standard curriculum-learning strategy in modern LM RL training, namely pass-rate filtering that removes prompts that do not contribute useful gradient signal (Section 2.3). The empirical results show that RLVE is consistently better in both learning efficiency and final model performance.
> > > + In **Section 3.3**, we explain in detail how each environment is designed so that solving all higher-difficulty problems strictly subsumes solving any lower-difficulty ones from the same environment.
> > > + In the rebuttal, we also ran an additional experiment. Specifically, when training Qwen2.5-7B-Base on the Sorting environment, we track the lowest difficulty level at which accuracy falls below 90%. If this value decreases over training, that would indicate forgetting of lower-difficulty problems. In practice, [the curve generally monotonically increases, suggesting that forgetting is not an issue here](https://ibb.co/3513QysT).
> > >
> > > We believe these analyses support both the motivation for RLVE’s design and the role of its key components. **If the reviewer has a particular missing analysis in mind, we would sincerely appreciate that clarification.**

---

### Official Review · Reviewer_8uLv · 2026-03-12

**Soundness:** 3
**Presentation:** 3
**Significance:** 3
**Originality:** 3
**Overall Recommendation:** 4
**Confidence:** 4

**Summary:**

This paper proposes the RLVE (Reinforcement Learning with Verifiable Environments) framework, which aims to scale reinforcement learning for language model training through adaptive verifiable environments. The core idea is to construct a benchmark suite called RLVE-GYM, consisting of 400 verifiable environments that can programmatically generate tasks and provide deterministic verification feedback, thereby addressing the sparse and unreliable reward signals commonly encountered in reinforcement learning for language models.

From a technical perspective, RLVE adopts a training paradigm similar to DeepSeek-R1 and uses the GRPO (Group Relative Policy Optimization) algorithm. Experiments are conducted on models ranging from 1.5B to 7B parameters. The evaluation covers multiple domains, including mathematical reasoning (AIME 2024/2025), code generation (LiveCodeBench v6), logic puzzles (OMEGA-500, OlympiadBench), and complex reasoning (BBE). The paper particularly emphasizes the adaptive capability of the environments, which can dynamically adjust task difficulty according to the model’s current performance, enabling a curriculum learning effect. The training infrastructure uses a single-node configuration with 8×H100 GPUs, supports response lengths up to 24,576 tokens, and each training run requires approximately 2–8 days (about 350–1,500 GPU hours).

**Compliance With Llm Reviewing Policy:**

Affirmed.

**Key Questions For Authors:**

See Weaknesses

**Limitations:**

yes

**Strengths And Weaknesses:**

**Strengths**

1.	The paper addresses the issue of diminishing learning signals caused by static RLVR data distributions and proposes replacing fixed datasets with adaptively adjusted difficulty, enabling dynamically controlled training data and more scalable RL training.
2.	The experimental design is comprehensive, comparing different models and RL training settings across multiple scenarios, and the results demonstrate the advantages of the RLVE framework.
3.	The RLVE framework covers multiple domains, including mathematics and code, indicating good generality and scalability.

**Weaknesses and Questions**
1.	What are the main differences between RLVE and existing carefully designed curriculum learning strategies? In smaller-scale training settings (e.g., on a specific dataset), does RLVE still provide advantages over traditional static RLVR approaches?
2.	RLVE dynamically increases task difficulty as training progresses. However, if the model’s performance temporarily degrades during training, is it possible for the environment to reduce the difficulty accordingly?

---

> ### Author Rebuttal · Authors · 2026-03-27
>
> We sincerely thank Reviewer 8uLv for the thoughtful review. We appreciate the recognition of RLVE’s significance, experimental thoroughness, and effectiveness. Below, we address the concerns in turn.
>
> + Reviewer 8uLv asks about the main differences between RLVE and existing curriculum learning strategies.
>   + **We discuss this distinction in the second paragraph of Appendix A**. We will move this discussion up to Related Work in the camera-ready version.
>   + Specifically, curriculum learning has long been used to improve RL efficiency by presenting training data in order of increasing difficulty, and similar ideas have recently been extended to LMs. However, such approaches typically operate on a finite dataset by reordering existing problems post hoc, whereas RLVE defines difficulty levels over an effectively infinite problem space induced by each verifiable environment and trains by progressively advancing through each level. In other words, instead of estimating difficulty and reordering a fixed dataset, RLVE uses environments with predefined, controllable difficulty.
>   + Importantly, each environment in our work can generate **an effectively infinite number of problems with unbounded difficulty**. As a result, **it is impractical to first materialize such problems as a finite dataset and then apply existing curriculum-learning methods**. RLVE allows us to keep learning progressively rather than being capped by a fixed difficulty ceiling.
>   + In fact, **Section 4.1 already compares RLVE against a bounded-difficulty setup combined with an existing curriculum-learning strategy**. Specifically, we always use dynamic prompt filtering (Section 2.3), which has become a standard curriculum-learning practice in modern LM RL training, as it keeps only prompts at appropriate difficulty levels that can provide useful gradient signals. The empirical results show that RLVE is consistently better.
>
> + Reviewer 8uLv asks whether RLVE can be applied in settings based on a specific dataset.
>   + Based on the discussions above, RLVE cannot be directly applied to an arbitrary fixed dataset since it must operate on environments as defined in Section 2.1.
>   + That said, although RLVE does not train directly on fixed real-world datasets, we show that RLVE-GYM helps models acquire reasoning capabilities that generalize beyond the training environments. We show from our results in Section 5 that RLVE significantly improves performance on reasoning-intensive real-world benchmarks (AIME, LiveCodeBench, BBEH, etc).
>
> + Reviewer 8uLv asks whether, although RLVE dynamically increases task difficulty as training progresses, the environment can also reduce the difficulty if the model’s performance temporarily degrades during training.
>   + We agree this could be necessary if forgetting of easier problems occurs. However, as discussed in Section 3.3, we design each environment so that **solving all higher-difficulty problems strictly subsumes solving lower-difficulty ones from the same environment. This makes forgetting theoretically limited.**
>   + We also ran an additional experiment to support this. Specifically, when training Qwen2.5-7B-Base on the Sorting environment, we track the lowest difficulty level at which accuracy falls below 90%. If this value decreases over training, that would indicate forgetting. In practice, the [curve](https://ibb.co/3513QysT) generally monotonically increases, suggesting that forgetting is not an issue here.

---

> > ### Author Rebuttal · Reviewer_8uLv · 2026-04-01
> >
> > The rebuttal clarified the distinction between RLVE and standard curriculum learning more convincingly, especially by emphasizing controllable difficulty over effectively infinite problem spaces and by pointing to comparisons against bounded-difficulty curriculum baselines already included in the paper. This addresses an important part of my concern.
> > However, the response on whether difficulty can decrease when model performance regresses is less fully resolved: the additional evidence is helpful, but remains limited to a specific environment/model setting and does not demonstrate a general adaptive downward-adjustment mechanism.
> > Overall, the rebuttal improves my understanding of the paper and partially addresses my concerns, but does not substantially change my overall assessment.

---

> > > ### Author Response · Authors · 2026-04-01
> > >
> > > Thank you very much for the follow-up. We are glad that our rebuttal clarified the distinction between RLVE and standard curriculum learning.
> > >
> > > Below, we further address **the only remaining concern regarding the absence of an explicit downward-adjustment mechanism** when model performance temporarily degrades during training.
> > >
> > > + Our main reason for not adding such a mechanism is that, as discussed in Section 3.3, we design each environment so that **solving all higher-difficulty problems strictly subsumes solving any lower-difficulty ones from the same environment. This makes forgetting theoretically limited.**
> > >   + Examples
> > >     + In Sorting, a higher difficulty means a longer array. If a model can correctly sort arrays of length $N+1$, it must also be able to sort arrays of length $N$, since the latter are strict subproblems of the former. Concretely, given any length-$N$ array, we can construct a length-$(N+1)$ array by inserting a new smallest element; solving the longer array immediately yields the sorted order of the original one.
> > >     + In Integration, a higher difficulty means a larger expression tree. Solving all integrals with $N+1$ nodes presupposes the ability to solve those with $N$ nodes. Concretely, given any integrand $f$ with $N$ nodes, we can construct a harder $(N+1)$-node instance such as $f+1$; solving $\int (f+1)\,dx$ immediately yields $\int f\,dx$ by subtracting $x$.
> > >     + In Polynomial Minimization, a higher difficulty means a higher polynomial degree. Solving minimization problems for all polynomials up to degree $N$ presupposes the ability to solve those of lower degrees. Concretely, given any polynomial $p(x)$ of degree $N-1$, we can construct a degree-$N$ polynomial $q(x)=p(x)+0\cdot x^N$, which is formally harder under our difficulty definition but has exactly the same minimizer as $p(x)$.
> > >   + We design all environments to follow this same principle. Therefore, if the model can solve all harder problems in an environment, it should also retain the ability to solve all easier ones from that environment. This is why we do not introduce a separate downgrading mechanism.
> > >
> > > + A second reason is algorithmic simplicity and robustness.
> > >   + In typical RL training, performance naturally fluctuates, so it is difficult to reliably distinguish meaningful regression from ordinary noise. This becomes even harder in our setting because each difficulty level could correspond to an effectively infinite set of problems, making performance estimation inherently hard.
> > >   + Rather than introducing additional complexity through explicit regression detection and downward adjustment, we use the sliding difficulty window described in Section 2.2. Concretely, even after the right endpoint moves to harder levels, training samples are still drawn from a range of difficulties, so the model continues to see lower-difficulty problems as training progresses.
> > >
> > > + Finally, our ultimate goal is not to maximize performance on the constructed environments themselves, but to improve general reasoning on real-world tasks. Empirically, the current design already leads to strong downstream gains on benchmarks such as AIME, LiveCodeBench, and BBEH (Section 5), suggesting that an explicit downward-adjustment mechanism is not necessary in practice for the settings studied here.
> > >
> > > + Separately,  we also realized that the new experimental result may have been easy to miss in our rebuttal due to the low contrast between the text color and the URL color, so we highlight it again here: [https://ibb.co/3513QysT](https://ibb.co/3513QysT). Sorry for any potential confusion!
> > >
> > > We hope this clarifies why RLVE does not currently include explicit difficulty reduction, and why we believe the present design is both theoretically well motivated and practically sufficient.

---

### Official Review · Reviewer_DeMD · 2026-03-13

**Soundness:** 3
**Presentation:** 3
**Significance:** 4
**Originality:** 4
**Overall Recommendation:** 4
**Confidence:** 4

**Summary:**

This work proposes RLVE, a new paradigm that scales LLM RL in terms of scaling verifiable environments. The authors manually craft 400 environments with automatic programs for result verification and train LLMs with synthesized questions with dynamic difficulties. Through analysis the authors validate the necessity of dynamic difficulty and scaling environments, and full-scale experiments show that RLVE enables LLMs to both surpass RL saturation and improve RL training within a limited computation budget.

**Compliance With Llm Reviewing Policy:**

Affirmed.

**Final Justification:**

I thank the authors for their efforts in rebuttal, and I will maintain my positive score.

**Key Questions For Authors:**

1. RLVE requires LLMs to manually produce the correct outputs without executing code. However, there are many extremely hard problems that LLMs can hardly learn to manage like NP-complete problems, and in many real-world scenarios there are useful tools that LLMs can use to simplify their task. Therefore, I feel that scaling only reasoning tasks is somehow a bit limited, and I wonder whether the authors have considered or attempted building and scaling interactive environments.
2. A big issue with curriculum learning is forgetting. Since dynamic difficulty is used to progressively train LLMs towards harder tasks in higher sample-efficiency, I wonder whether easier tasks suffer from forgetting, and I feel some experiment results on this would be helpful.
3. In Figure 3, why does the effective prompt ratio first increases and then quickly drops?
4. According to Section 4.2, expanding training environments improves performance, can we recognize which one(s) of the environment is critical so that we can only involve a small proportion of environments at each different training stage and for different downstream tasks? This might reduce the cost in further scaling up as this work has involved 400 environments.
5. Dynamic difficulty is closely related to curriculum learning, and it is implemented as a sliding range without explanation. I feel additional experiments would be helpful to understand the implementation choices and their corresponding effect.
6. The authors mention the high cost of building DeepMath-103K ($138,000 USD and 127,000 GPU hours), and I feel explicitly stating the cost in building the 400 environments of RLVE would make the comparison more intuitive.
7. Section 5.2, Line 389: The meaning of this sentence is not clear to me: "Practically, these experiments also simulate the standard LM post-training pipeline, where LMs are first trained via SFT and subsequently via RL". What does "these experiments" refer to and why it simulates the SFT+RL pipeline?

**Limitations:**

Yes.

**Strengths And Weaknesses:**

**Strengths**

1. RLVE is a novel paradigm that scales RL training in a new dimension, which may inspire a series of follow-up works.

2. The analyses justify the effectiveness of dynamic difficulty and environment scaling, and the results from training on all environments validate that RLVE enables further improvement over saturation.

3. The idea is generally well motivated and explained in the paper.

**Weaknesses**

1. RLVE is not compatible with open-ended generation that are not verifiable and currently does not involve agentic tasks.

2. RLVE is currently implemented by manual environment engineering, which is costly.

3. Presentation of some details is not sufficiently clear. See questions below.

---

> ### Author Rebuttal · Authors · 2026-03-27
>
> We sincerely thank Reviewer DeMD for the thoughtful review. We appreciate the recognition of RLVE’s novelty, effectiveness, and overall clarity. Below, we address the concerns in turn.
>
> + Reviewer DeMD notes that RLVE currently focuses on verifiable, non-agentic environments, and suggests scaling interactive/agentic environments.
>   + We agree this is a promising future direction, and **in Section 7, we already discuss both the importance of building non-verifiable environments and the associated challenges**.
>    + As a **first step** toward RL with adaptive environments in the research community, we focus on reasoning-centric environments because reasoning is a core capability underlying many useful real-world tasks; our results do show that **the trained models perform well on challenging real-world tasks** (AIME, LiveCodeBench, BBEH, etc).
>   + Our environments also provide a relatively clean and lightweight research testbed. While agentic environments are also important, **academic research often benefits from settings that are easier to study and reproduce**, without requiring heavy infrastructure such as complex Docker-based execution.
>
> + Reviewer DeMD raises the cost of manual environment engineering.
>   + Our contribution is both the RLVE methodology and the RLVE-GYM environment suite. **RLVE itself is compatible with both manually or automatically constructed environments**.
>   + **We also discuss this in Section 7**. In brief, we explored automatic environment construction with frontier models, but found it difficult to maintain sufficiently high environment quality without human intervention. We therefore consider the authors’ manual effort worthwhile as a strong prototype for future work on automatic environment generation; **rather than viewing this merely as a limitation, we believe RLVE-GYM itself is a valuable artifact and an important contribution to the community**.
>   + We view manual engineering as a practical way at the present time, not as a necessary long-term requirement. Analogously, training data were once largely hand-curated, whereas synthetic data are now common. Given the rapid progress of frontier LMs in coding, we expect the scalability limitation of manual environment construction should not remain a major obstacle.
>
> + Reviewer DeMD asks whether dynamic difficulty may cause forgetting of easier problems.
>   + We agree this could happen when the difficulty is only loosely defined. However, **as discussed in Section 3.3, we design each environment so that solving all higher-difficulty problems strictly subsumes solving lower-difficulty ones from the same environment. This makes forgetting theoretically limited**.
>   + Following Reviewer DeMD’s suggestion, we also ran an additional experiment. Specifically, when training Qwen2.5-7B-Base on the Sorting environment, we track the lowest difficulty level at which accuracy falls below 90%. If this value decreases over training, that would indicate forgetting. In practice, the [curve](https://ibb.co/3513QysT) monotonically increases, suggesting that forgetting is not an issue here.
>
> + Reviewer DeMD asks why the effective prompt ratio in Figure 3 first increases and then quickly drops.
>   + This is due to a quite high threshold (90%) for advancing the difficulty.
>   + This conservative hyperparameter choice avoids increasing difficulty too aggressively and thus destabilizing training, but it also lowers the effective prompt ratio: for example, once accuracy at the current frontier reaches 70%–80%, many prompts are already too easy to provide useful learning signals.
>
> + Reviewer DeMD asks whether a subset of environments can be selected to match training on all environments.
>   + We fully agree that this is an important research direction. We believe techniques from data selection (e.g., influence functions) are promising candidates for environment selection.
>
> + Reviewer DeMD asks about the motivation for the sliding difficulty window.
>   + We explain this in the third paragraph of Section 2.2. The left endpoint is introduced to prevent the difficulty range from expanding indefinitely.
>
> + Reviewer DeMD asks for the cost of building RLVE-GYM.
>   + All environments were manually engineered by the paper authors, so there was no direct monetary cost.
>   + As a reference, building one environment took roughly 1 hour on average for a CS PhD student. As a rough upper-bound estimate, a CS Ph.D. student in the U.S. is typically paid less than 50USD per hour. The total human-labor cost of constructing RLVE-GYM is roughly bounded by 400×1×50=20,000USD.
>
> + Reviewer DeMD asks about the sentence in Section 5.2: “Practically, these experiments …”
>   + We apologize for the unclear wording. “These experiments” refers to the compute-efficient scaling experiments in Section 5.2.
>   + The experiments mirror the standard LM post-training pipeline of first applying SFT and then RL. Therefore, the results suggest RLVE is effective in realistic LM post-training settings.

---

> > ### Author Rebuttal · Reviewer_DeMD · 2026-04-02
> >
> > I thank the authors for their thorough rebuttal and my concerns are fully resolved. I will maintain my score. Good luck with your submission.

---

> > > ### Author Response · Authors · 2026-04-02
> > >
> > > Thanks very much! We are glad that all of your concerns have been addressed.

---

### Decision · Program_Chairs · 2026-04-30

**Decision:**

Accept (regular)

**Comment:**

This paper presents RLVE, a novel framework that scales reinforcement learning for large language models through the use of adaptive verifiable environments. By dynamically adjusting the difficulty of tasks to match the model's improving capabilities, RLVE moves away from static datasets to provide continuous, informative learning signals during training.

Overall, reviewers found the paper to be technically robust and well-executed, praising both its conceptual framing and empirical validation. They specifically highlighted the novelty of scaling RL via environment diversity, noting consistent improvements across various reasoning benchmarks. Furthermore, the creation of a large dataset of verifiable environments is widely seen as a significant contribution that will spur future work.

The authors' rebuttal was highly effective. Two reviewers felt their concerns were completely addressed, while the remaining issues were either partially resolved or deemed to require experiments beyond the current scope. Crucially, reviewers identified no major flaws in the paper's methodology, concluding that its central claims are well-supported by the evidence.